# Massively parallel reporter assays and mouse transgenic assays provide correlated and complementary information about neuronal enhancer activity

Michael Kosicki [1,9], Dianne Laboy Cintrón[2,3,9], Pia Keukeleire [4], Max Schubach [5], Nicholas F. Page[2,3,6], Ilias Georgakopoulos-Soares [7], Jennifer A. Akiyama [1], Ingrid Plajzer-Frick[1], Catherine S. Novak[1], Momoe Kato[1], Riana D. Hunter[1], Kianna von Maydell[1], Sarah Barton[1], Patrick Godfrey[1], Erik Beckman[1], Stephan J. Sanders [3,6,8], Martin Kircher[4,5], Len A. Pennacchio [1] & Nadav Ahituv [2,3] ✉

High-throughput massively parallel reporter assays (MPRAs) and phenotype-rich in vivo transgenic mouse assays are two potentially complementary ways to study the impact of noncoding variants associated with psychiatric diseases. Here, we investigate the utility of combining these assays. Specifically, we carry out an MPRA in induced human neurons on over 50,000 sequences derived from fetal neuronal ATAC-seq datasets and enhancers validated in mouse assays. We also test the impact of over 20,000 variants, including synthetic mutations and 167 common variants associated with psychiatric disorders. We find a strong and specific correlation between MPRA and mouse neuronal enhancer activity. Four out of five tested variants with significant MPRA effects affected neuronal enhancer activity in mouse embryos. Mouse assays also reveal pleiotropic variant effects that could not be observed in MPRA. Our work provides a catalog of functional neuronal enhancers and variant effects and highlights the effectiveness of combining MPRAs and mouse transgenic assays.

Genome-wide association studies (GWAS) have identified hundreds of non-coding variants associated with psychiatric disorders, which exhibit complex genetic etiologies likely involving multiple loci[1–6]. The GWAS-discovered lead variants are not necessarily causative due to linkage disequilibrium (LD), which increases the number of potential variant candidates on average by ten-fold. In addition, ongoing whole genome sequencing studies of patients with psychiatric disorders identify ~70 de novo non-coding variants per individual[7]. These efforts

[1]Environmental Genomics & Systems Biology Division, Lawrence Berkeley National Laboratory, Berkeley, CA 94720, USA. [2]Department of Bioengineering and Therapeutic Sciences, University of California San Francisco, San Francisco, CA 94158, USA. [3]Institute for Human Genetics, University of California San Francisco, San Francisco, CA 94158, USA. [4]Institute of Human Genetics, University Medical Center Schleswig-Holstein, University of Lübeck, 23562 Lübeck, Germany. [5]Berlin Institute of Health at Charité – Universitätsmedizin Berlin, 10117 Berlin, Germany. [6]Department of Psychiatry and Behavioral Sciences, Kavli Institute for Fundamental Neuroscience, Weill Institute for Neurosciences, University of California, San Francisco, San Francisco, CA, USA. [7]Institute for Personalized Medicine, Department of Biochemistry and Molecular Biology, The Pennsylvania State University College of Medicine, Hershey, PA 17033, USA. [8]Institute of Developmental and Regenerative Medicine, Department of Paediatrics, University of Oxford, Oxford OX3 16 7TY, UK. [9]These authors contributed equally: Michael Kosicki, Dianne Laboy Cintrón. ✉e-mail: nadav.ahituv@ucsf.edu

highlight the challenge to distinguish causative variants from the hundreds of thousands of potential candidates identified through genetic studies.

Various genomic correlates of function can be used to reduce the number of potential candidates. Putative regulatory sequences can be identified in a tissue and even cell-type specific manner using such methods as DNase-seq and ATAC-seq (identifying open chromatin regions), or ChIP-seq[8–12] (identifying regions bound by transcription factors or having specific histone marks). Variants falling in regulatory regions with activity in relevant cell types are more likely to be causative. However, an overlap between a variant and regulatory region neither confirms variant functionality, nor provides a mechanism for how it impacts the phenotype. Functional assays that can test the effect of the variant on gene regulatory activity are needed to pinpoint the causative mutations.

Massively parallel reporter assays (MPRA) allow for the assessment of regulatory activity of tens of thousands to hundreds of thousands of candidate regulatory sequences and variants within them[13–15]. The majority of MPRAs are conducted in vitro, enabling the high throughput interrogation of candidate sequences and variants in a quantitative and reproducible manner. However, they are limited to testing the function of the assayed sequence only in the specific cell type and cannot assess how results relate to its function in vivo. As an alternative, in vivo activity of human enhancers can be tested using a transgenic mouse assay (referred to as "transgenic assay" below) such as enSERT[16,17]. In this assay, a candidate regulatory sequence is coupled to a minimal promoter and reporter gene followed by its integration into a safe harbor locus in mouse zygotes and assayed for activity by imaging at a later embryonic time point. Transgenic assays can identify enhancer expression at an organismal level, providing rich, multitissue phenotype. Results of thousands of these assays are cataloged in the VISTA enhancer browser and serve as a gold standard for enhancer activity assessment[18]. However, these assays are more resource and labor intensive than MPRAs and therefore are typically conducted at a much lower throughput.

Combining the high throughput capabilities of MPRAs and rich phenotype of transgenic assays is an underexplored venue for regulatory element and variant characterization. Limited comparisons of these technologies have been performed[19–23], typically involving MPRAs conducted in cancer or immortalized cell lines with limited relevance to organismal biology, use short sequences (120 bp) or sampled too sparsely from in vivo validated sequences to enable a systematic comparison.

Here, we set out to robustly compare results between MPRA and transgenic assays by using psychiatric disorders-associated sequences and variants as a test case. We carry out an MPRA for over 50,000 sequences 270 bp in length, many of which were derived from brain enhancers in the VISTA enhancer browser[18] and over 20,000 variants. We find thousands of functional regulatory sequences and hundreds of variants that alter regulatory activity compared to their reference allele. We observe an overall strong correlation between MPRA and transgenic assays. Variants with a high impact in MPRA also had a significant effect on neuronal enhancer activity in transgenic assays in mouse embryos. Combined, our work provides a large catalog of functional neuronal enhancers and their variants, and shows that MPRAs can be successfully combined with mouse transgenic enhancer assays.

## Results
### MPRA neuronal library composition and initial QC
We set out to investigate the correlation between high-throughput MPRAs and mouse enhancer transgenic assays. As neuronal enhancers are the most abundant category of enhancers in the VISTA Enhancer Browser[18], which catalogs the results of mouse transgenic assays conducted on human and mouse sequences, and since our lab has

established MPRA protocols in stem cell differentiated human neurons[19,24,25], we focused on neuronal-associated elements. We designed an MPRA library by tiling peaks from five single-cell and bulk neuronal ATAC-seq experiments[26–30] and from conserved cores of 1400 neuronal and non-neuronal enhancers from the VISTA Enhancer Browser[18] with 270 bp tiles (Fig. 1a, b; minimum 30 bp overlap; see Methods). To characterize how mutations affect the activity of these elements, we introduced two types of variants into the designed tiles. First, we included all lead single-nucleotide polymorphisms (SNPs) and SNPs in linkage disequilibrium ($r^2 > 0.8$) from autism spectrum disorders (ASD), schizophrenia, bipolar disorders and depression GWAS that overlapped designed tiles[1,3–5]. Second, we introduced synthetic transversion variants into every fourth base pair of elements with high likelihood of MPRA activity (overlapping ATAC-seq peaks from multiple datasets, evolutionary conserved, active in transgenic assay, see Methods; Fig. 1a, b)[31]. As negative controls, we used 500 di-nucleotide scrambled, non-conserved tiles from enhancers negative in mouse transgenic assays that did not have overlapping ENCODE candidate cis-regulatory elements[32] or neuronal ATAC-seq signal[26–30]. In total, we designed 81,952 unique 270 bp sequences, including 24,942 variants.

Oligos were synthesized and cloned into a barcoded lentiMPRA vector and packaged into lentivirus following our previously published protocol[15]. They were then transduced into differentiated human excitatory neurons derived from an isogenic WTC11-Ngn2 iPSC line with an inducible Neurogenin-2 gene using an established induction protocol[15,33,34]. Only tiles with at least 15 barcodes detected in each of the three replicates were retained (mean = 103 barcodes post-filtering; Supplementary Fig. 1a) and tiles with mutations without a reference tile passing these criteria were discarded. Out of 81,952 elements, 73,367 passed QC (90%; see Methods), including 50,083 genomic elements, 22,710 single base pair mutation tiles and 454 scrambled negative controls. Together, the elements covered 11.3 Mbp of genomic sequence in 23,961 non-overlapping regions of 270 bp (tile size) to 6531 bp in size (mean 472 bp). MPRA activity was expressed as a z-score of log2(RNA counts/DNA counts) relative to scrambled negative controls. Negative control reference tiles, which were selected from non-conserved parts of elements negative in transgenic assay and with no epigenomic signal in neural datasets, had a similar activity to their scrambled counterparts (Supplementary Fig. 1b, c). This both validated their selection strategy and showed that scrambling did not systematically make elements active or repressive. We observed good correlation between replicates (Pearson correlation = 0.76–0.78, Supplementary Fig. 1b, N = 73,367). Using a multiple-testing corrected p-value < 0.05 relative to the 2.5th–97.5th percentile interval (-+/−2 standard deviations) of the scrambled negative controls, we designated 742 tiles to be activators and 732 tiles to be repressors (out of 50,083, 2.9%). Using similar criteria, we found 454 single base pair mutation tiles to have decreased activity compared to reference tile and 315 to have increased activity (out of 22,710, 3.4%).

### MPRA captures neuronal-specific activity
To validate the results of our MPRA, we annotated the activity of tiles overlapping a variety of genomic annotations. Specifically, we asked if ranks of the overlapping tiles were significantly different from scrambled negative controls (Fig. 2a; Supplementary Data 1). On average, elements in our library were more active than scrambled negative controls (median activity = 0.09). Overlap with positive elements in previous neuronal MPRAs was associated with higher activity, with elements from Inoue 2019[19] (double-Smad inhibition protocol) being more active than those from Uebbing 2021[35] (stable neural stem cell line, median activity 0.16 vs 0.11). We also confirmed that tiles that were pre-selected for mutagenesis due to high expected activity were indeed highly active ("Mutation reference tiles", activity = 0.17). At the positive extreme, tiles overlapping housekeeping promoters (defined as 2 kb centered around the 5′ end of Gencode protein-coding exon 1

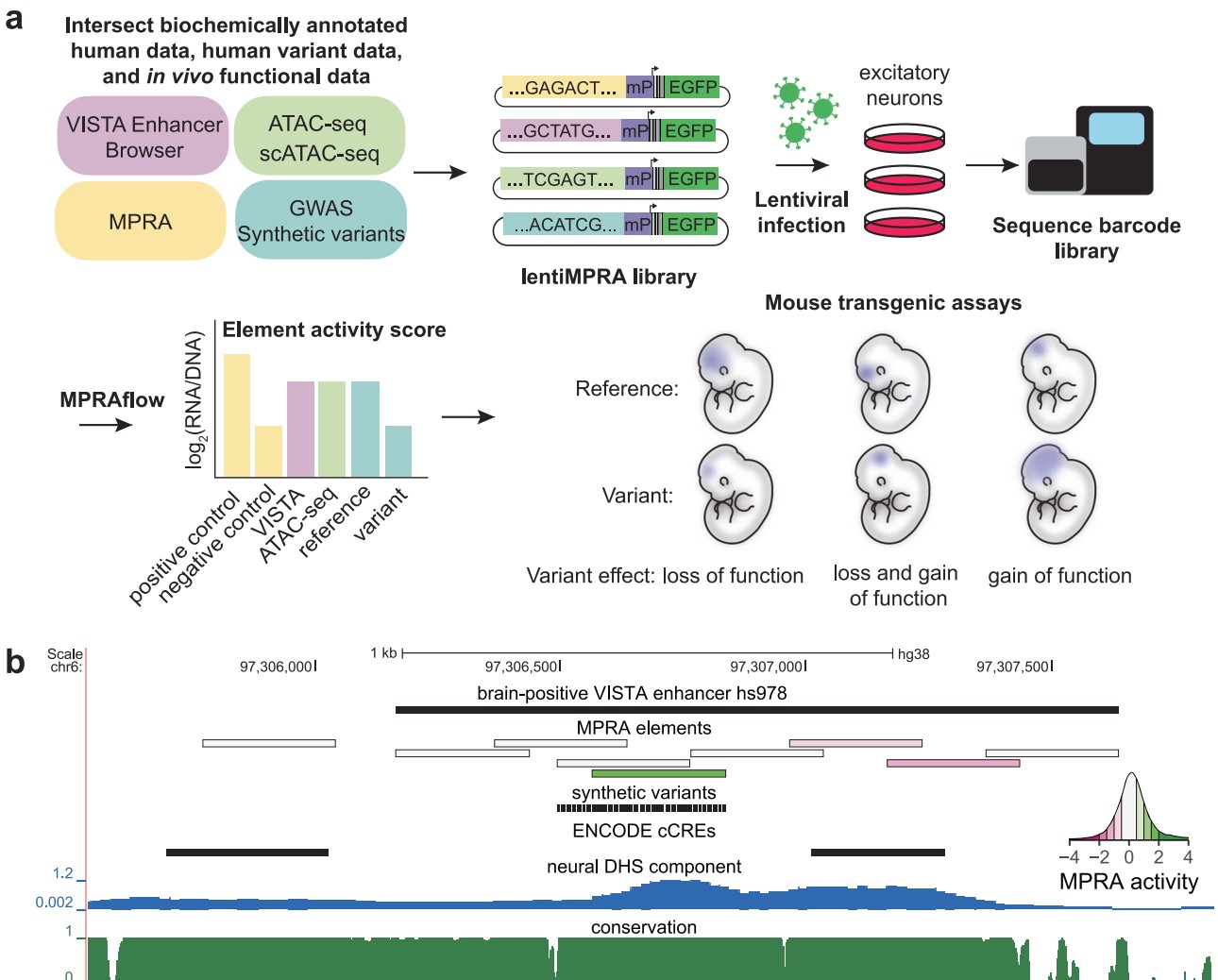

**Fig. 1 | Functional validation of candidate *cis*-regulatory elements (cCREs) using lentiMPRA and mouse transgenic assays. a** Schematic of experimental plan. A lentiMPRA library was designed through the intersection of scATAC-seq, ATAC-seq, VISTA Enhancer Browser[18], conservation and neuronal MPRA data. The library also included GWAS lead SNPs and SNPs in LD with them and synthetic variants. Sequences were inserted into a reporter plasmid upstream of a minimal promoter (mP), barcode and EGFP. The library was infected into WTC11 induced excitatory neurons using lentivirus. The integrated DNA and transcribed RNA barcodes were sequenced to determine element activity scores. Mouse transgenic assays were conducted on selected sequences to characterize their activity in vivo. **b** UCSC Browser annotation, from top to bottom: (1) VISTA enhancer browser hs978 sequence (2) MPRA elements colored by MPRA activity with green showing high activity and pink low activity (see inset). (3) synthetic variants included in MPRA tested elements (4) ENCODE cCRE (candidate *cis*-regulatory elements)[32] (5) Neural DNase I hypersensitivity signal component[76] (6) PhastCons conservation UCSC track for 30 mammals (27 primates).

of genes in Eisenberg and Levanon 2013[36]) were highly active (median activity 0.32), suggesting that they can be used as universal positive controls in MPRAs. We note that they may function as autonomous promoters and not as enhancers[37]. Ultraconserved elements[38] had high activity as well (0.19), which is consistent with our previous observation that they are often active in the developing brain in transgenic assays[39]. Conversely, tiles overlapping coding exons, but not exons of long-non-coding RNAs, were overall repressive (median activity = −0.15). It is unlikely splicing sites present at exon-intron interface explain this observation, as in our MPRA the minimal promoter, and consequently the transcription start site, are downstream of the tested element.

We then set out to analyze the transcription factor binding sites (TFBS) to further validate that our MPRA captures neuron-specific signals. Using HOMER[40], we compared activator or repressor tiles that do not overlap promoters ($N = 309$ and 335) to either genomic background or to other tiles from our MPRA with background level activity ("scramble-like"; $−0.4 <$activity$< 0.4$, $N = 20,974$; Fig. 2b; Supplementary

Data 2). As input, we used HOMER mouse and human TFBSs ($N = 439$). The analysis accounted for GC-content differences in test and background sets. We considered a TFBS to be enriched if it was present in at least 15% of test tiles, increased by at least two times compared to background set (corresponding to $\log_2(2) = 1$ cutoff) and was significantly enriched by HOMER's hypergeometric test at FDR-adjusted $p$-value $< 0.01$. Raw analysis files are provided as Supplementary Data 1.

We found a total of 30 motifs to be enriched in activator tiles compared to either genomic background or tiles with scramble-like activity. These included neuron-associated motif families such as RFX, LHX, CUX, ELK and DLX[41–43], nervous system expressed SP5 and HNF6 motifs[44] as well as growth/survival TFs from the ATF family. This demonstrates our MPRA captured neuron-specific signal (Supplementary Fig. 2a–c).

Repressor tiles were enriched for Nkx6.1 and four motifs from the SOX family[45,46], but only in analysis using genomic background. This may imply lack of specific repressive signal in our library, limited power to detect such signal or relative dearth of known repressive

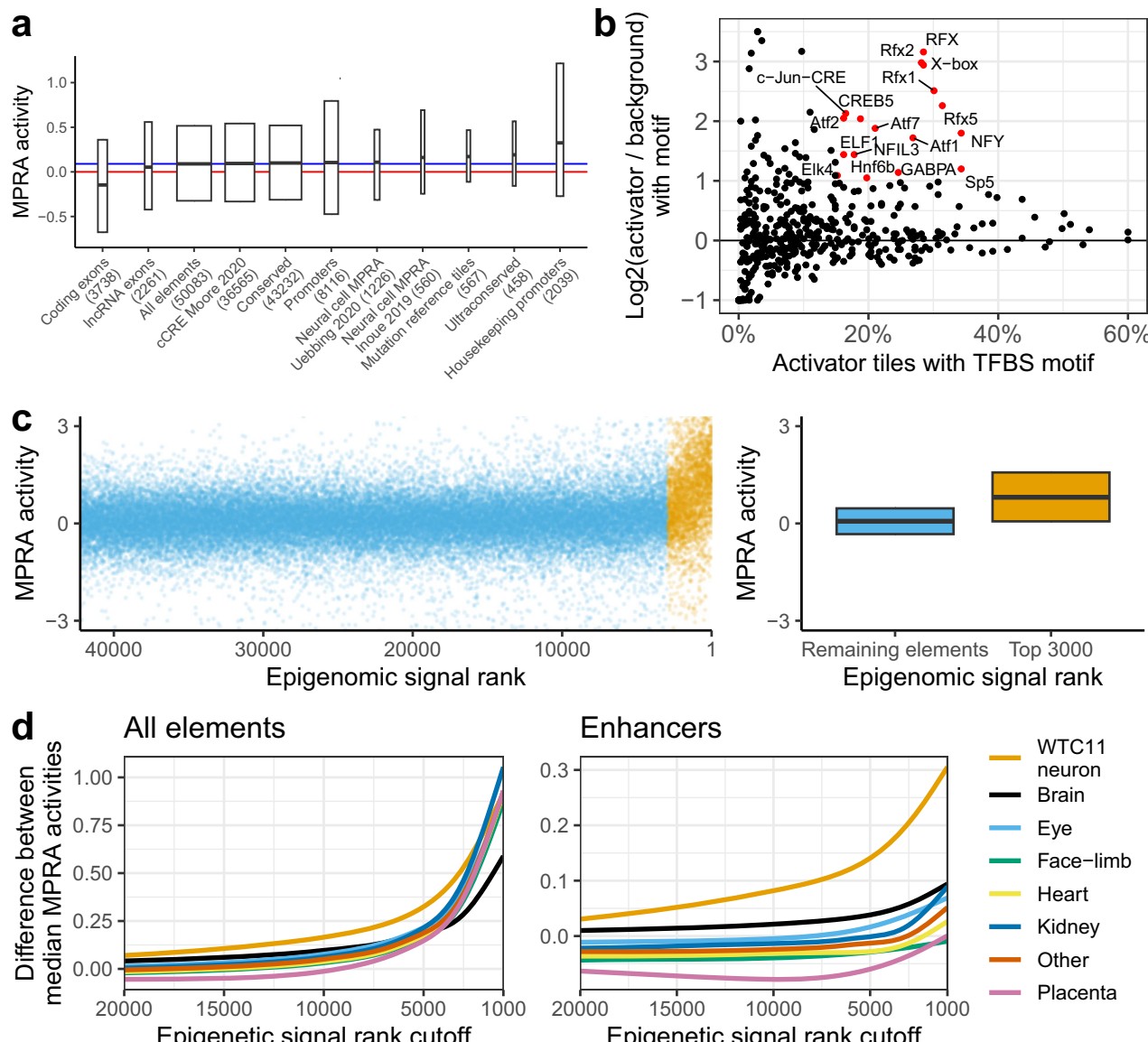

**Fig. 2 | Neuronal WTC11 MPRA results validation. a** MPRA activity of tiles overlapping different categories. Red line = activity of scrambled negative controls (zero, by definition). Blue line = median activity of all reference elements (0.09). Hinges of boxplot span interquartile range, line in the middle is median, width is proportional to the number of overlapping tiles. All categories except lncRNA exons had significantly different activity than scrambled negatives at an FDR-adjusted *p*-value < 0.05 (Mann-Whitney U test). Promoters are defined as the 5′ end of exon 1 of protein-coding genes +/−1 kb. **b** TFBS enrichment in enhancer, activator tiles compared to enhancer elements with scramble negative levels of activity. Log2-fold change was curbed at −1 and 3.5. Fraction of tiles with motif was curbed at 60%. Only TFBSs present in >15% of targets, with two times increase in presence

from background to target set (corresponding to log2(2) = 1 cutoff) and FDR < 1% are labeled and colored red. **c** Methodology for comparison of epigenomic annotation. Left: tiles were ranked by epigenomic signal and split at various rank cutoffs into two groups. Right: Median MPRA activity of the two groups was compared. *N* = 39,260 for "Remaining elements". Same boxplot display conventions as in (**a**). **d** Difference in median MPRA activity at different epigenomic rank cutoffs for eight tissue groups. Left: all elements (*N* = 42,260; lower than 50,083 total due to exclusion of elements that failed to lift-over between human and mouse genomic assemblies), right: enhancers (*N* = 36,248; enhancers defined as not overlapping "coding promoter" category in (**a**); see Methods).

motifs in the the HOMER dataset. The latter might be consistent with de novo motif analysis conducted against tiles with background activity, which revealed similar motifs for activator tiles (RFX7/Rfxdc2, X-box, SP5, match scores 0.82–0.9), but novel repressor motifs with tentative matches to ZIC2, KLF15, and SP1 among others (match scores of 0.7–0.75; Supplementary Data 3).

We observed that both activator and repressor tiles had higher median levels of GC-content than the rest of the library, with repressors having higher levels than activators (repressors 74%, activators 55%, remaining elements 45%; Supplementary Fig. 2d). Such GC-skew should not affect MPRA readout on a technical level, as the activity of

the tested element is read through sequencing of a barcode, not the element itself (unlike e.g. in STARR-seq). We conclude that highly GC-rich sequences may function as transcriptional repressor elements in this MPRA.

We next set out to assess how well various biochemical marks correlated with neuronal MPRA activity. To investigate whether activity signal in our MPRA is biologically specific, we compared our MPRA results to epigenomic signal from diverse tissues and cell types from 12 embryonic, fetal and WTC11 datasets, encompassing 740 DNase hypersensitive sites (DHS), ATAC and single-cell ATAC samples (Supplementary Data 4). To account for a large diversity of experimental

and computational protocols, we integrated raw genomic signal (bigWig tracks) over MPRA tiles and ranked the tiles based on the signal for each dataset. We then computed the difference between median MPRA activity of top ranked elements and the remaining ones for a range of epigenomic rank cutoffs (Fig. 2c). As expected, the more stringent the rank cutoff, the larger the difference between activity of top ranked elements versus the rest. However, due to enrichment of promoter-overlapping elements in top ranks, the differences between individual datasets were negligible (Fig. 2d, left). After removing tiles overlapping protein-coding promoters, we observed a clear separation of brain and differentiated WTC11 cells samples from non-neuronal samples (Fig. 2d, right). Closer inspection revealed that some non-neuronal ENCODE DHS samples (adrenal, eye and kidney) are still enriched, especially at stringent cutoffs, possibly reflecting a combination of high activity tissue-invariant elements ("housekeeping enhancers") and higher signal-to-noise ratio of DHS data at high signal intensities. This was attenuated at less stringent signal cutoffs, with only two non-neuronal samples (retina and kidney) remaining in top 50 at signal rank cutoff 5000 (Supplementary Data 4). Encouragingly, we observed a clear separation over an ATAC-seq time course of WTC11 cell neuronal differentiation, with undifferentiated cells ranking at position 589, day 3 differentiated cells at position 61 and day 14 at position 12[47]. We note that our MPRA design sampled elements with open chromatin signal in neuronal tissues more deeply than in other tissues, which may have contributed to the observed enrichments. In summary, our results show that our MPRA captures neuronal-associated regulatory activity.

## Neuronal MPRA activity correlates with mouse neuronal enhancer expression

The average sequence length tested in transgenic mouse assays is around 1 kb, about four times the size of tiles in our MPRA (270 bp). To compare these two assays, we matched transgenic assay elements ("VISTA elements") with overlapping MPRA tiles of highest activity (Fig. 3a). We then built a general linear model (GLM) with a binomial link to predict binary, tissue-specific transgenic assay results (e.g. brain activity, yes or no) from MPRA activity. In our design, we have included negative control tiles derived from non-conserved parts of negative VISTA elements that did not overlap epigenomic signal from any of the neural datasets. Conversely, we aimed to capture as many conserved parts of positive VISTA elements as possible (Fig. 3a). To account for this design bias, we included a fraction of conserved sequences covered by tiles as a covariate in the model (Fig. 3b, c). We found that all neural annotations (except dorsal root ganglion) were significantly correlated with MPRA activity, while craniofacial and heart terms were significantly negatively correlated (Fig. 3c). The steepest regression slope on the positive side was for a combined 'neural' term (brain, neural tube, cranial nerves including trigeminal nerve and dorsal root ganglion), followed by 'brain'. An alternative modeling solution, removing VISTA elements with poor coverage of conserved regions, yielded similar results (Supplementary Fig. 3a–c). In contrast, an uncontrolled model, while showing similar neuronal and heart term enrichments, also showed a strong negative correlation with negative VISTA elements, demonstrating the impact of uneven sampling of conserved regions in positive and negative elements (Supplementary Fig. 3d).

To validate the model, we selected 6 elements with an average predicted likelihood of in vivo neuronal activity of 86% and tested them using transgenic mouse assays. We found that 5 out of 6 such elements (83%) were active in neuronal tissues, showing that the model is well calibrated (Supplementary Fig. 4). We also validated our approach by building a model to predict transgenic assay results using data from an independent, published MPRA conducted in primary human fetal cortical cells[48]. Similar to our MPRA, forebrain and combined brain terms were positively correlated with MPRA activity, while

heart, heart+somite and facial-mesenchyme were negatively correlated (Supplementary Fig. 3e). We note that for this MPRA, a model taking into account sequence conservation across species was very similar to the model that did not. This likely reflects the fact that conservation signal was not used to select regions for testing in this MPRA (Supplementary Fig. 3e). We conclude that neural MPRA in differentiated human excitatory neurons and neural activity in transgenic assay strongly correlate in a tissue-specific manner.

## MPRA effect of psychiatric disorder associated GWAS variants

We next analyzed the 167 psychiatric disorder associated GWAS variants tested in our MPRA. We found 7 out of 167 variants (4.2%) had a significant effect on MPRA activity (9.4%; Supplementary Data 5; Supplementary Fig. 5). Five of the variants resulted in loss of activity, and two resulted in gain of activity, compared to the reference allele. Each variant was associated (in linkage disequilibrium) with one or multiple independent GWAS lead SNPs, covering a total of 11/117 psychiatric disorder associated GWAS loci (9.4%). Using our general linear model, we found that there was a 70% average probability that regulatory elements overlapping these variants will show activity in neuronal tissues in vivo. We also found that 5 out of 7 variants were either in close proximity to genes associated with neuronal biology, including *CTNND1*, *GRIN2A* and *MAU2*, or linked to them using the Activity-By-Contact (ABC) model (Supplementary Data 5)[49]. We conclude that our MPRA identified regulatory, neuropsychiatric-disorder associated variants with plausible impact on neuronal biology.

## Variants altering MPRA activity affect neuronal mouse enhancer activity

To select variants for transgenic assay follow-up, we analyzed the synthetic, single nucleotide variants. Out of 20,126 successfully tested non-GWAS variants (within 22,710 variant tiles), 751 had a significant effect on regulatory activity (FDR-corrected *p*-value < 0.05). We selected five of these variants for follow up in the transgenic assay, based on prior evidence of neuronal activity in transgenic assays and, to a lesser degree, links to important neuronal genes predicted using the ABC model (e.g. *QKI*, *PRKN*, *COA7* and *MEF2C*; Table 1, Fig. 4a)[49].

We found that 4 out of 5 variants affected mouse enhancer expression in a reproducible manner, in the direction consistent with the MPRA impact. Three variants caused a loss of activity in different parts of the brain, neural tube or cranial nerves, and the fourth variant caused overall gain of neural activity (Fig. 4b). In one case, predicted loss of activity was accompanied by a gain of expression in another brain-associated structure. We note that the one variant with no apparent impact on transgenic enhancer activity had a very high basal activity of the reference element in the transgenic assay (hs268), which may have masked expression differences due to the variant. These results demonstrate the utility of combining the two experimental systems, with a good correspondence between MPRA and mouse transgenic assay and rich additional information provided by the latter.

To further interpret the results of our transgenic assays, we used motifbreakR[50] to carry out TFBS predictions for the five variants tested using a transgenic assay (Fig. 4c, d; Supplementary Fig. 7). In four cases, more than one plausible TFBS was found. We leveraged the fact our MPRA design also contained variants in close proximity to the ones we selected for transgenic assay follow up to further validate the TFBS predictions. For example, the variant tested in the hs978.1 element was predicted to both destroy the POU4F3 site and create a potential CDX1 site, but the MPRA effects of variants overlapping the TFBS were more consistent with POU4F3 destruction than with CDX1 creation (Fig. 4c). We applied similar logic to remaining predictions to select the most plausible of the initial TFBS matches (see Fig. 4d for TFBSs consistent with overlapping variant MPRA effects and Supplementary Fig. 7 for rejected TFBSs). Deploying this approach in a systematic manner could help interpret future variant MPRAs.

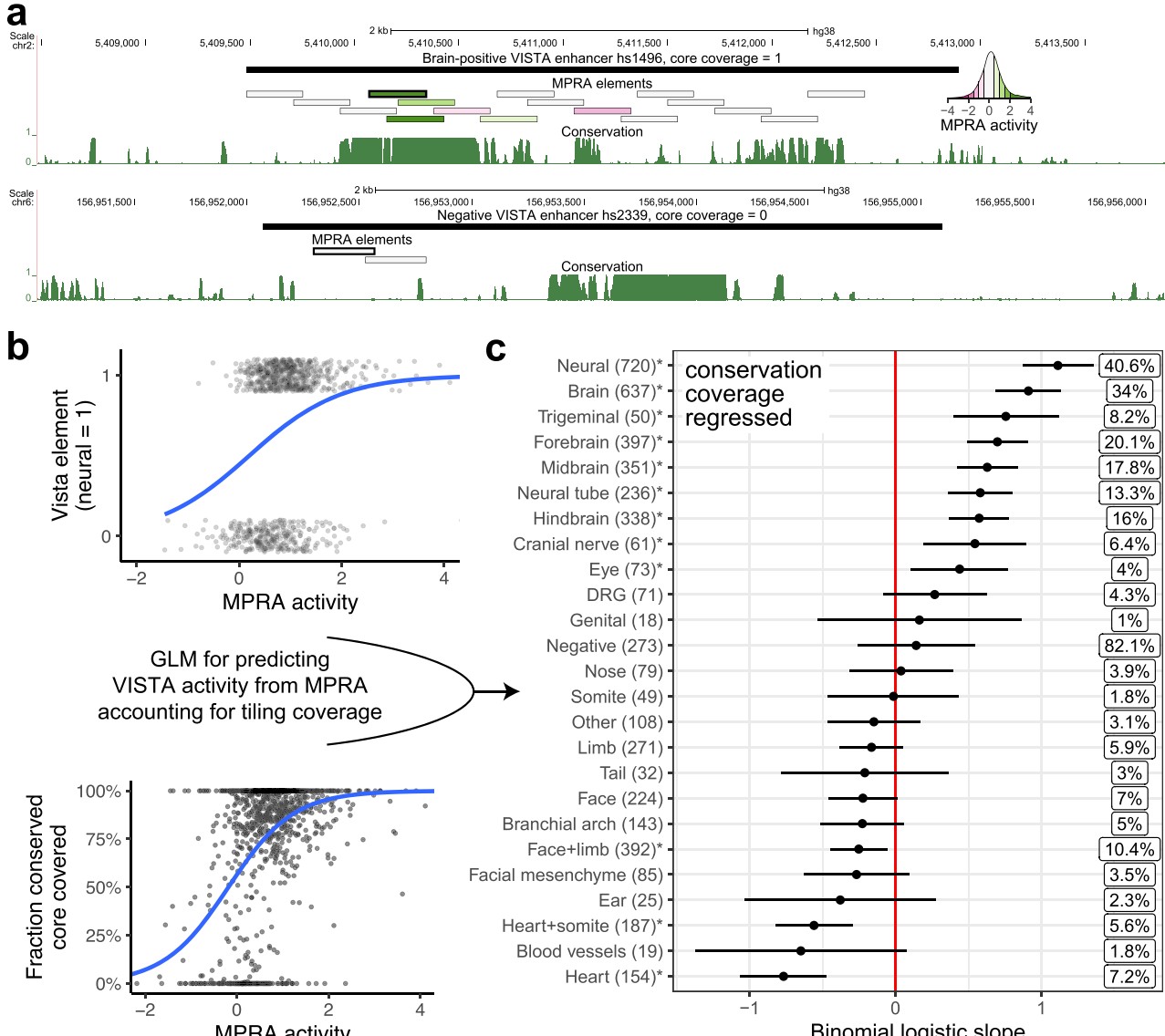

**Fig. 3 | Predicting transgenic assay activity using an MPRA-based, coverage-corrected model. a** Examples of VISTA elements with complete (top) and zero (bottom) coverage of their sequence conserved cores using MPRA tiles. Conservation is shown as the PhastCons UCSC track for 30 mammals (27 primates). The MPRA tile with highest activity (used for modeling) has a thicker border. MPRA elements are colored by MPRA activity, see inset. **b** Visualization of input variables for the GLM. Top: transgenic assay (VISTA) elements are binarized according to chosen tissue activity (here: neural; jitter added for visualization). The blue line is the binomial-link GLM regression on this variable. Bottom: relationship between fraction of conserved core covered and MPRA activity is modeled as a covariate. The blue line is the binomial-link GLM regression on this variable. **c** Results of the GLM predicting binomial transgenic assay activity from MPRA activity and fraction of conserved core covered. Asterisks indicate $p$-value < 0.05 (likelihood ratio test, no multiple testing correction). Boxed percentages to the right are Nagelkerke $R^2$ measures. Bars extend two standard errors of the mean in each direction. DRG = dorsal root ganglia. Face-mesen = facial mesenchyme. Cranial nerves category does not include the trigeminal nerve, as per VISTA Browser.

## Discussion

We performed an MPRA in neurons with elements derived from VISTA enhancers and neuronal fetal ATAC-seq peaks finding a good correlation to neuronal expression in mouse transgenic assays. In terms of variants, we did not see a strong effect on MPRA activity for our selected psychiatric disorder associated GWAS variants, but observed effects on MPRA activity for 783 out of 20,126 synthetic variant tiles. Four out of five synthetic variants nominated by MPRA as having a significant impact also affected the transgenic assay activity in the expected manner and revealed additional ectopic effects. Overall, we demonstrated that combining MPRA and transgenic assays can be highly advantageous.

The observed complementarity of the two assays is encouraging. MPRAs allow the testing of a large number of sequences and provide a quantitative readout, while transgenic assays can reveal the organismal spatially-resolved consequences of regulatory sequences and variants. Both approaches have improved significantly over the past decade, coming closer to bridging the gap between them. MPRAs have been increasing in throughput, length of tested elements, range of cell types amenable to this type of assay (due to lentivirus and AAV delivery) and have also been carried out in vivo in select tissues in a postnatal manner[15,22,23,51–53]. Transgenic assays improved in throughput and reproducibility due to the development of Cas9-guided safe harbor integration method enSERT[17]. However, both methods still suffer from

**Table 1 | MPRA variants tested in transgenic mouse assay**

| Variant name | Reference MPRA activity | Variant MPRA effect | TFBS affected | VISTA element | Structures affected in transgenic assay | Neural target genes | Relevant target gene phenotypic associations |
|---|---|---|---|---|---|---|---|
| chr6-162856979-C-G | 2 | −1.9 | RORB (gain), **CTCF** | hs2793.1 | partial forebrain loss | QKI, PACRG and PRKN | Associated with Parkinson's Disease (PACRG, PRKN) and schizophrenia (QKI) |
| chr6-97306759-A-T | 1.9 | −2.2 | CDX1 (gain), **POU4F1** | hs978.1 | partial forebrain, hindbrain and neural tube loss | GPR63 | Branchiooculofacial Syndrome and Spina Bifida Occulta |
| chr1-52663061-C-G | 3.1 | −2.4 | **SP2** | hs2790.1* | midbrain gain, partial forebrain loss | COA7, TUT4 and others | Spinocerebellar Ataxia (COA7) Perlman Syndrome (TUT4) |
| chr13-80276542-C-G | 0.2 | +6.7 | **NFY (gain)** | hs3117.1* | midbrain and hindbrain gain | SPRY2 | Decreased brain expression in schizophrenia and bipolar disorder |
| chr5-88396638-C-G | −0.1 | −1.3 | ASCL1 (gain), **NR2F1** | hs268.1 | no effect | MEF2C | Associated with cognitive disability, epilepsy and cerebral malformations |

TFBS predictions from motifbreakR. Motifs consistent with variant effects in bold (see Fig. 4c). Neural target genes found using activity-by-contact (ABC) on data from WTC11 excitatory neurons (cell line used in this study) or prenatal week 18 prefrontal cortex neurons (Methods).
*Element not previously tested in the transgenic assay.

significant shortcomings. MPRAs conducted in vitro are limited to the cell types in which they are assayed, can be limited by the availability of differentiation protocols and labor intensity of differentiating millions of cells with various identities and cannot assess the spatial and temporal organismal activity of a regulatory element. enSERT is conducted in mice, which cannot capture all aspects of human biology, is costly and not high-throughput. It also has only recently been applied in a quantitative manner[54]. While both methods are likely to improve and eventually merge, our work highlights the utility of combining currently available approaches, with MPRA as a high-throughput filter for the multi-tissue transgenic assay. We note that we only used lentiMPRA here and other MPRA technologies could potentially lead to different results though they are usually well correlated[37].

We observed a significant effect on MPRA activity for 7 out of 167 tested psychiatric disorder associated GWAS variants (4.2%), corresponding to 11/117 GWAS loci (9.4%). This is in line with another MPRA carried out by our lab that found 164 psychiatric disorder and eQTL variants out of 17,069 tested (<1%) to have an effect on MPRA activity[53]. It is unclear how many of the GWAS signals are driven by regulatory elements, but it is possible we missed regulatory variants with lower effect sizes, associated with non-transcriptional phenotypes, like chromatin tethering[55], or variants having an effect in another cell type or at a different differentiation time point. We note that machine learning models of MPRA data and saturation mutagenesis experiments[31,53] show that rare variants have a higher effect on MPRA activity compared to common variants.

Synthetic variants comprised the majority of variants tested in this MPRA, which has some advantages over testing common variants. First, the effect sizes of these variants are not constrained by negative selection, unlike common variation in human populations. This makes synthetic MPRA a better substrate for computational modeling, which should be able to learn a wide range of potential effects. Second, our experiment allowed us to find functional variants in elements likely to control expression of neuronal genes, some of which are linked to neurodevelopmental disorders. These results place a strong prior on interpretation of yet undiscovered, large effect de novo variants in these regions and can help better understand the regulatory biology of neuronal development.

In summary, we compiled a catalog of transcriptional activity in neuronal cells of over 50,000 elements derived from open chromatin fetal datasets and enhancers validated in transgenic assays. We also assessed the impact of over 20,000 synthetic and 167 GWAS variants, and demonstrated the usefulness of using MPRA as a variant filter for transgenic mouse assays. We anticipate this work will contribute to computational modeling of gene regulation and studies focused on neural development and psychiatric disorders.

## Methods

### Ethical statement

All animal work was reviewed and approved by the LBNL Animal Welfare and Research Committee. All iPSC experiments were conducted following UCSF institutional guidelines. No commonly misidentified cell lines were used in the study.

### MPRA design

We used following datasets for our library design - VISTA enhancers, MPRA tiles from Inoue 2019[19] (activity >1.1 at both 48 h and 72 h timepoints) and Uebbing 2021[35] ($q < 0.05$ for both replicates, following the publication) and single-cell or bulk ATAC or ATAC-seq peaks called by Ziffra 2020[26] (26,000 peaks designated enhancers by activity-by-contact), Domcke 2020[27] (33,000 cerebrum peaks with mean expression > 0.1), Preissl 2018[28] (top 15,000 peaks from each of eEX1, eIN1, eIN2 and RG1-4 clusters), Gorkin 2020[29] (top 15,000 from forebrain, midbrain, hindbrain and neural tube e11.5 samples), Inoue 2019[19] (top 15,000 peaks from 72 h timepoint) and Song 2019[30] (WTC11 neurons;

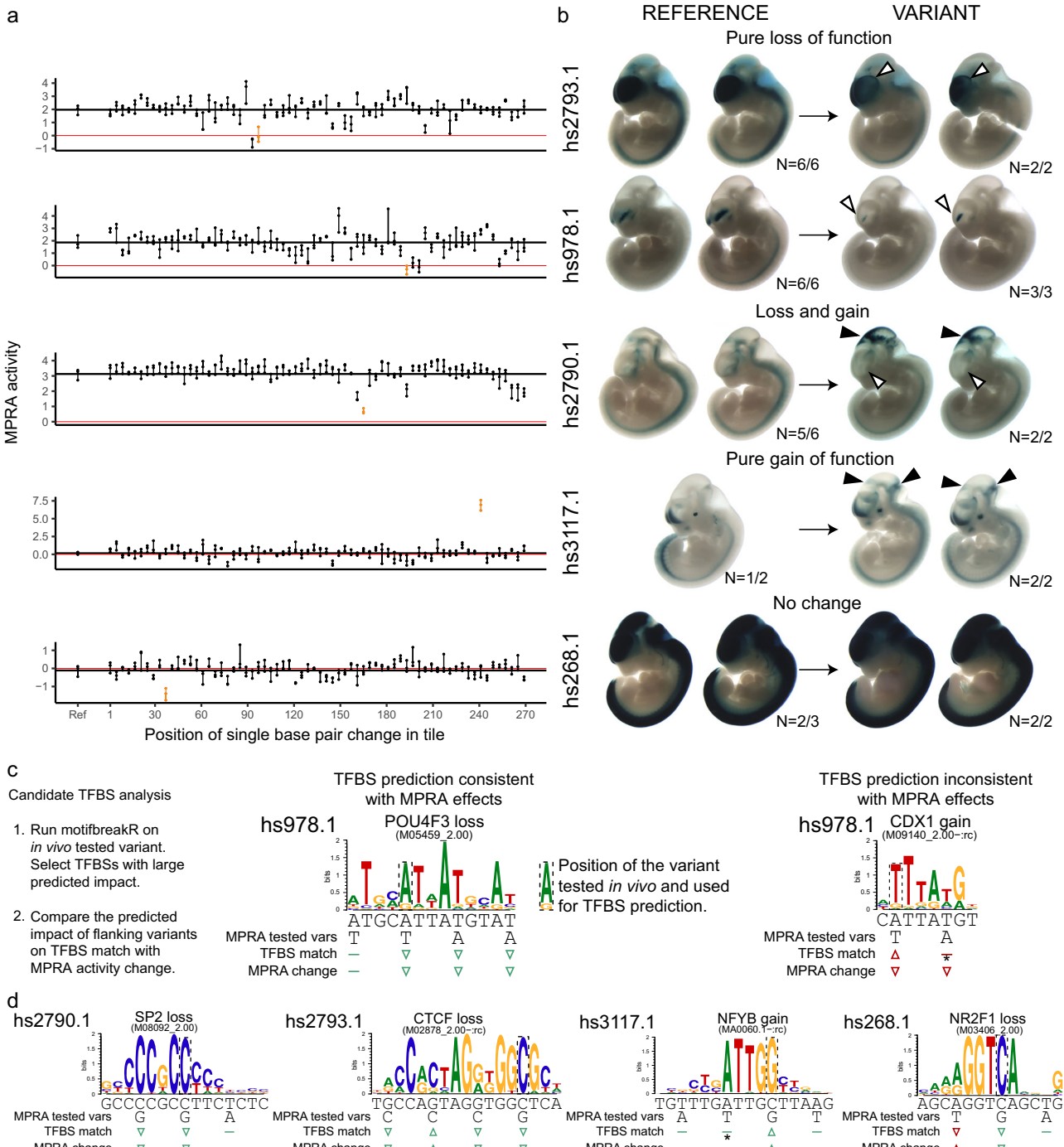

**Fig. 4 | Synthetic MPRA variants lead to in vivo change of function in transgenic assay. a** Every fourth nucleotide of five MPRA tiles was mutagenized individually, for a total of 67 mutagenized constructs per reference tile. Dots connected by a vertical line = three biological replicates. Red horizontal line = zero, mean activity of scramble negative controls. Black horizontal line = mean activity of the reference construct. **b** Constructs encompassing the MPRA tiles, with or without the variants indicated in orange in panel A, were tested for activity using transgenic assay in developing mouse embryos at embryonic day e11.5. All variants except hs268.1 led to change of function in one or more neural tissues - brain, neural tube or cranial nerves. Shown are embryos that were genotyped as "tandem", i.e. positive for insertion at the safe harbor locus and presence of the plasmid backbone indicating multi-copy insertion with strong, reproducible pattern. White arrowhead indicates loss of function, black indicates gain of function. See

Supplementary Fig. 6 for all embryo images, including those genotyped as "tandem" and "single" (see Methods), which provide additional support of the changes observed. **c** An example of TFBS predicted by motifbreakR to be affected by the in vivo tested variants in hs978.1. Left: TFBS prediction consistent with all MPRA variant effects (POU4F3), right: TFBS prediction inconsistent with MPRA effects (CDX1 gain). Predicted TFBS match change and measured MPRA activity change symbols are colored green if matching and red if not. Arrowheads indicate an increase or decrease, flat line indicates no effect (assuming predicted TFBSs are activating). **d** Prediction of TFBSs likely affected by the variants and partially or fully consistent with flanking variant effects. Asterisk - TFBS events that are predicted to be created by in vivo tested variants are assumed to be unaffected by loss-of-activity flanking variants i.e. no loss of binding is possible, where no binding was observed in the reference sequence.

top 15,000 peaks; Supplementary Data 6). Numbers of peaks/tiles covered in the final library are available in Supplementary Data 7. Elements were either extended to 270 bp (if shorter) or tiled in intervals of 270 bp with a minimum 30 bp overlap. We also designed tiles directly upstream of the first exons of coding genes in Gencode v34 with neural ATAC signal (one tile per promoter), facing in the direction of transcription and avoiding overlap with FANTOM5 CAGE TSS peaks[56]. Tiles centered on representative DHS elements with 'Neural', 'Organ devel. / renal' and 'Primitive / embryonic' annotation were added, if overlapping previously chosen elements[32]. Genomic negative control tiles (N = 500) were selected from sections of negative VISTA elements that were not conserved, not active in previous neural MPRAs and did not overlap any cCREs[32] or any of the peaks in ATAC datasets mentioned above. Finally, we used a weighted combination of evolutionary conservation (UCSC phastConsElements30way[57]), lack of overlap with coding exons (Gencode[58]) and promoters regions (Gencode and CAGE[56]), neural VISTA activity, presence of a peak in multiple ATAC datasets, activity in previous neural MPRAs[19,35] and overlap with LD blocks from psychiatric disorder GWAS to select 56,387 hg38 genomic reference tiles. We used 500 genomic negative control tiles to form di-nucleotide scramble negative controls. All resources that were not originally available in hg38 (including mouse VISTA enhancers), were lifted over using Kent tools and relevant UCSC chain files[59]. The design was conducted in R 4.3.2 with tidyverse 2.0.0 package[60,61]. Additional information about the used data and software can be found in Supplementary Data 6.

We introduced mutations into 595 reference tiles, resulting in 123 tiles with multiple SNVs (derived from random mutagenesis of ultra-conserved VISTA elements[62]) and 24,942 with individual SNVs. To select GWAS SNPs for testing, we started with a set of 465 common lead SNPs from psychiatric disorder GWAS[1,3–5] and extracted 15,133 SNPs in linkage disequilibrium (LD) with these using SNiPA ($r^2 > 0.8$, 1000 genomes set, v3). We selected 186 for testing based on overlap with enhancers with high likelihood of activity (overlapping ATAC-seq peaks from multiple datasets, evolutionary conserved, active in transgenic assay or highly active in previous neuronal MPRAs). To investigate vulnerabilities of regulatory elements associated with GWAS signals, we conducted systematic mutagenesis of every fourth nucleotide in 85 tiles within 0.8 LD regions for additional 5621 SNVs using a GC-preserving transversion scheme (G = C, A = T). Finally, we conducted similar systematic mutagenesis of 254 tiles with high likelihood of activity for an additional 17,272 GC-preserving transversion SNVs. Numbers of SNVs listed above are mutually exclusive, but some SNVs belonged to more than one category. For example, a total of 5,892 SNVs were in 0.8 LD regions, including synthetic, lead and LD SNPs. Note that about 10% of all designed elements were not successfully tested - see Results section for numbers after QC.

## LentiMPRA cloning and infection

The lentiMPRA library was constructed as previously described[15]. A synthesized TWIST oligo pool with 300 bp long elements (270 bp insert + 2*15 bp PCR handles) was amplified by 12-cycle PCR using NEBNext High-Fidelity 2x PCR Master Mix (New England BioLabs, M0541L), the forward primer 5BC-AG-f01 and reverse primer 5BC-AG-r01 were used to add the minimal promoter, spacer and vector overhang sequence. The amplified fragment was purified using 1x of the HighPrep PCR Clean-up System (Magbio, AC-60500). The purified fragment underwent a second round of 12-cycle PCR using NEBNext High-Fidelity 2x PCR Master Mix (New England BioLabs, M0541L), the forward primer 5BC-AG-f02 and the reverse primer 5BC-AG-r02. This step added a 15 bp random barcode downstream from the minimal promoter. The amplified fragment was purified using Nucleospin Gel and PCR-Clean-Up (Macherey-Nagel, 740609.50) and 1.2x HighPrep PCR Clean-up System (Magbio, AC-60500). The oligo library was cloned into the double digested AgeI/SbfI pLS-SceI vector (Addgene,137725) using the

NEBuilder HiFi DNA Assembly Master Mix (New England BioLabs, E2621L). The plasmid lentiMPRA library was electroporated into MegaX DH10B T1R Electrocomp Cells (Invitrogen, C640003) using the Gemini X2 (2.0 kV, 200 Ω, 25 µF). The electroporated cells were then plated on eleven 15 cm 100 mg/mL ampicillin LB agar plates (Teknova, L5004) and grown overnight at 37 °C. Approximately 8 million colonies were pooled and midi-prepped (Qiagen, 12145) to obtain on average 100 barcodes per oligo. To associate barcodes with each oligo in the library, the Illumina flow cell adapters were added through a 15-cycle PCR using NEBNext High-Fidelity 2x PCR Master Mix (New England BioLabs, M0541L), the forward primer pLSmP-ass-i741 and reverse primer pLSmP-ass-gfp. The amplified fragment was purified using Nucleospin Gel and PCR-Clean-Up (Macherey-Nagel, 740609.50) and 1.8x HighPrep PCR Clean-up System (Magbio, AC-60500). The amplified fragments were sequenced on a Illumina NovoSeq 500 using a NextSeq 150PE kit with custom primers (R1: pLSmP-ass-seq-R1, R2: pLSmP-ass-seq-ind1, R3: pLSmP-ass-seq-R2).

Lentivirus production was conducted on twenty-nine 10 cm dishes of LentiX 293 T cell line (TakaraBio, 632180) with Lenti-Pac HIV expression packaging kit (GeneCopoeia, LT002) following the manufacturer's protocol. Lentivirus was filtered through a .45 µm PES filter system (Thermo Fisher Scientific,165-0045) and concentrated with Lenti-X Concentrator (TakaraBio, 631232). Titration of the lentiMPRA library was conducted on differentiated human excitatory neurons. Cells were seeded at $4.5 \times 10^4$ cells per well in a 12-well plate on day 0 and incubated for 7 days. Serial volumes of the lentivirus (0, 1, 2, 4, 8, 16, 32, 64,128 µL) were added along with 6 µL ViroMag R/L (OZ Biosciences, RL41000) per well. After lentivirus addition cells were incubated for 30 min on the magnet at 37°C. The magnet was removed and cells were incubated at 37°C for 7 days, the media was replaced after 24 h of lentivirus addition. The cells were washed with DPBS (Sigma-Aldrich, D8537) and DNA was extracted with the AllPrep DNA/RNA Mini Kit (Qiagen, 80204) following the manufacturer's protocol for DNA extraction. The multiplicity of infection (MOI) was determined as the relative amount of viral DNA over that of genomic DNA by qPCR using SsoFast EvaGreen Supermix (Bio-Rad, 1725205). The lentivirus infection, DNA/RNA extraction and DNA/RNA barcode sequencing were conducted as previously described[15]. Each replicate required approximately 25 million cells. Therefore, cells were seeded at day 0 of differentiation in four 10 cm plates with 5 million cells each. On day 7, the cells were infected with the lentivirus library and ViroMag R/L (OZ Biosciences, RL41000) following the manufacturer's protocol. All three replicates were infected with the same lentivirus batch at an MOI of 80. Media was replaced 24 h after lentivirus addition and the cells were incubated for 7 days. DNA and RNA were extracted from the three replicates using the AllPrep DNA/RNA Mini Kit (Qiagen, 80204) following the manufacturer's protocol. The RNA was treated with the TURBO DNA-free Kit (Life Technologies, AM1907) following the manufacturer's protocol for rigorous DNase treatment. Reverse transcription was conducted with SuperScript II Reverse Transcriptase (Life Technologies, 18064-071) using a barcode-specific primer (P7-pLSmP-ass16UMI-gfp) which contains a 16 bp UMI. After DNAse treatment and reverse transcription the resulting cDNA and extracted DNA underwent the same steps to prepare the library for sequencing. To add a sample index and UMI, DNA and cDNA from the three replicates were kept separate and underwent a 3-cycle PCR using NEBNext Ultra II Q5 Master Mix (New England Biolabs, M0544L), forward primer P7-pLSmp-ass16UMI-gfp and reverse primer P5-pLSmP-5bc-i#. Another round of PCR was conducted to prepare the library for sequencing using NEBNext Ultra II Q5 Master Mix (New England Biolabs, M0544L), forward primer P5 and reverse primer P7. The fragments were purified using 1.2x of the HighPrep PCR Clean-up System (Magbio, AC-60500). The final libraries were sequenced on four runs of Illumina NextSeq high-output using the custom primers (R1: pLSmP-ass-seq-ind1, R2: pLSmP-UMI-seq, R3: pLSmP-bc-seq, R4: pLSmP-5bc-seq-R2).

## Cell culture and neuronal differentiation

Differentiated human excitatory neurons were derived from hiPSCs in the WTC11 background where a doxycycline-inducible neurogenin 2 transgene was integrated into the AAVS1 locus[33]. This cell line was generated by Dr. Li Gan's lab at the Gladstone Institutes from the parental WTC11 line. The WTC11 cell line was generated with donor consent at the Gladstone Institutes by Dr. Bruce Conklin's laboratory. We obtained the cells from the lab of Dr. Yin Shen at the University of California, San Francisco (UCSF). In the undifferentiated stage, cells were maintained in mTeSR 1 (STEMCELL Technologies, 85850) and the medium was changed daily. Once confluent, cells were washed with 1x DPBS (Sigma-Aldrich, D853), dissociated with accutase (STEMCELL Technologies, 07920) and plated a at 1:6 ratio in matrigel (Corning, 354277) coated plates. Media was supplemented with ROCK inhibitor Y-27632 (STEM CELL, 72304) at 10 μM on the day of passage. To initiate differentiation, cells were washed with 1x DPBS, dissociated with accutase and plated in matrigel-coated plates. For 3 days cells were cultured in KnockOut DMEM/F−12 (Life Technologies, 12660-012) medium supplemented with 2 μg/mL doxycycline (Sigma-Aldrich, D9891), 1X N-2 Supplement (Life Technologies, 17502-048), 1X NEAA (Life Technologies, 11140-050), 10 ng/mL BDNF (PeproTech, 450-02), 10 ng/mL NT-3 (PeproTech, 450-03) and 1 μg/mL lamininin (Life Technologies, 23017-015). The pre-differentiation medium was changed daily for three days and on the first day medium was supplemented with 10 μM ROCK inhibitor Y-27632. To induce neuronal maturation, cells were lifted and plated in Poly-L-Ornithine (Sigma-Aldrich, P3655) coated plates. Cells were cultured in maturation media containing Neurobasal A (Life Technologies, 12349-015) and DMEM/F12, HEPES (Life Technologies, 11330-032) with 2 μg/mL doxycycline supplemented with 1X N-2 Supplement, 0.5X B-27 Supplement, 1X NEAA, 0.5X GlutaMax (Life Technologies, 35050-061), 10 ng/mL BDNF, 10 ng/mL NT-3 and 1 μg/mL lamininin. A half-media change was conducted on day 7 and day 14 of differentiation using the maturation medium minus doxycycline.

## LentiMPRA analysis

Processing of barcode association and final MPRA libraries was done using a standardized MPRAflow pipeline[15,34,37], without a MAPQ filter to avoid artificial dropout due to multi-mapping of elements with single base pair mutations. All subsequent analyses were conducted in R 4.3.2 with tidyverse 2.0.0 package[61]. Barcodes with standard deviation of normalized log2(RNA/DNA) across replicates of > 1 were excluded from further analysis. BCalm 0.99.0 was used to further process the data, with total count normalization (instead of default normalization), and to call statistical significance for individual tiles and variants[63–65]. Visualizations were done using ggrastr 1.0.2 (https://github.com/VPetukhov/ggrastr), ggplot2 3.5.0[66] and ggrepel 0.9.5[67]. General linear models were constructed using rms 6.8−0 (https://hbiostat.org/r/rms/). Motifs affected by variants tested in the transgenic assay were detected using motifbreakR 2.16.0[50] with filterp=T, threshold=1e-4 and pwmList from Viestra 2020[68]. Only tiles with at least 15 barcodes detected in each of the three replicates were retained and mutation tiles without a reference passing these criteria were discarded as well. As per MPRAflow pipeline, these barcodes include ones detected in DNA or RNA. In other words, barcodes detected using only one modality were not discarded. MPRA activity was expressed as a z-score of log2(RNA counts/ DNA counts) relative to scramble negative controls.

## Correlation of MPRA activity and epigenomic signal

Epigenomic signal in the form of bigWig files was retrieved from ENCODE[32] and 12 other sources (Supplementary Data 5) and integrated over tile intervals using *bedtools bigWigAverageOverBed* command[69]. For each sample, signal was sorted and ranked with random tie-

breaking. For a range of rank cutoffs starting with 1000, the tiles were split into those above and below the cutoff and median MPRA activity was computed for both groups. The median activity of bottom signal group (e.g. from rank 1001 to lowest rank) was then subtracted from median activity of the top signal group (e.g. ranks 1–1000). For enhancer analysis, 8495 tiles overlapping promoters defined as 2 kb centered on the 5′ end of exon 1 of protein-coding genes in Gencode V34[58], were removed before computing the ranks and median activity difference.

## TFBS enrichment analysis

All analysis was done using HOMER 4.11[40] using activator or repressor tiles as target (as defined in the main text) and either HOMER-selected, GC-matched background genomic elements of the same size, or library elements with scramble negative levels of activity (−0.4 <activity <0.4). Only tiles not overlapping promoters, as defined in the previous section, were used. Default set of 239 unique TF motifs was used. Command of the form *findMotifsGenome.pl target.bed hg38 target_folder -bg background.bed -size 270 -nomotif* was run for each analysis, except -bg term was dropped for HOMER-selected background.

## Alignment and preprocessing of functional genomic data for ABC score pipeline

Gestational week 18 (GW18) bulk ATAC-seq and H3K27ac ChIP-seq data from human fetal prefrontal cortex[70] were aligned to hg19 using the standard Encode Consortium ATAC-seq and ChIP-seq pipelines respectively with default settings and pseudo replicate generation turned off (https://github.com/ENCODE-DCC). Trimmed, sorted, duplicate and chrM removed ATAC-seq and sorted, duplicate removed ChIP-seq bam files produced by the Encode pipeline were provided as input for calculating ABC scores.

ATAC-seq and H3K27ac CUT&RUN data from 7 to 8 week old NGN2-iPSC inducible excitatory neurons was obtained from Song 2019[30]. ATAC-seq and CUT&RUN reads were trimmed to 50 bp using TrimGalore[71] with the command --hardtrim 5 50 before alignment. ATAC-seq reads were aligned to hg19 using the standard Encode Consortium ATAC-seq and ChIP-seq pipelines respectively with default settings and pseudo replicate generation turned off. Trimmed, sorted, duplicate and chrM removed ATAC-seq bam files from multiple biological replicates were combined into a single bam file using samtools merge v1.10[72]. Trimmed CUT&RUN reads were aligned to hg19 using Bowtie2 v2.3.5.1[73] with the following settings --local --very-sensitive-local --no-mixed --no-discordant -I 10 -X 700 and output sam files were convert to bam format using samtools view[72,73]. Duplicated reads were removed from the CUT&RUN bam file using Picard MarkDuplicates v2.26.0[74] with the --REMOVE_DUPLICATES =true and --ASSUME_-SORTED=true options (http://broadinstitute.github.io/picard/). The final ATAC-seq and CUT&RUN bam files were provided as input for calculating ABC scores.

## Preprocessing of HiC and pcHiC data for ABC score pipeline

HiC contacts with 10 kb resolution from human GW17−18 fronto-parietal cortex was obtained in an hdf5 format separated by chromosome[75] (Supplementary Data 6). Hdf5 files were filtered for contacts with a score > 0 and converted into a bedpe format. Promoter capture HiC (pcHiC) contacts from 7 to 8 week old NGN2-iPSC inducible excitatory neurons were obtained in an ibed format from GSE113483[30]. The ibed file was converted to bedpe format and separated by chromosome. Bedpe files from GW17−18 cortex and iPSC derived excitatory neurons were provided as input for calculating ABC scores.

## Identification of candidate enhancer-gene pairs with ABC Score

The Activity-by-Contact (ABC) model identifies enhancer-gene relationships based on chromatin state and conformation[49]. Previously

identified open chromatin regions from GW18 human prefrontal cortex[70] and corresponding ATAC-seq and H3K27ac ChIP-seq bam files were provided as input for the ABC score pipeline MakeCandidateRegions.py script with the flags --peakExtendFromSummit 250 --nStrongestPeaks 150,000. Candidate enhancer regions identified were then provided to the run.neighborhoods.py script in addition to hg19 promoter merged transcript bounds. Finally, predict.py was used to identify final candidate enhancers using HiC data from human GW17−18 fronto-parietal cortex with the flags --hic_type bedpe --hic_resolution 10000 --scale_hic_using_powerlaw --threshold .02 --make_all_putative[75]. Candidate enhancer-gene pairs were also identified for 7−8p week old NGN2-iPSC inducible excitatory neurons using respective open chromatin regions[30], ATAC-seq and H3K27ac ChIP-seq data. All other settings for the ABC score pipeline remained constant. Promoter enhancer contacts from ABC score were saved as bedpe files and coordinates were mapped from hg19 to hg38 using liftOverBedpe (https://github.com/dphansti/liftOverBedpe).

### Mouse enhancer transgenic assay

Transgenic E11.5 mouse embryos were generated as described previously[16]. Briefly, super-ovulating female FVB mice were mated with FVB males and fertilized embryos were collected from the oviducts. Regulatory elements sequences were synthesized by Twist Biosciences. Inserts generated in this way were cloned into the donor plasmid containing minimal Shh promoter, lacZ reporter gene and H11 locus homology arms (Addgene, 139098) using NEBuilder HiFi DNA Assembly Mix (NEB, E2621). The sequence identity of donor plasmids was verified using long-read sequencing (Primordium). Plasmids are available upon request. A mixture of Cas9 protein (Alt-R SpCas9 Nuclease V3, IDT, Cat#1081058, final concentration 20 ng/μL), hybridized sgRNA against H11 locus (Alt-R CRISPR-Cas9 tracrRNA, IDT, cat#1072532 and Alt-R CRISPR-Cas9 locus targeting crRNA, gctgatggaacaggtaacaa, total final concentration 50 ng/μL) and donor plasmid (12.5 ng/μL) was injected into the pronucleus of donor FVB embryos. The efficiency of targeting and the gRNA selection process is described in detail in Osterwalder 2022[16]. Embryos were cultured in M16 with amino acids at 37°C, 5% CO2 for 2 h and implanted into pseudopregnant CD−1 mice. Embryos were collected at E11.5 for lacZ staining as described previously[16]. Briefly, embryos were dissected from the uterine horns, washed in cold PBS, fixed in 4% PFA for 30 min and washed three times in embryo wash buffer (2 mM MgCl2, 0.02% NP-40 and 0.01% deoxycholate in PBS at pH 7.3). They were subsequently stained overnight at room temperature in X-gal stain (4 mM potassium ferricyanide, 4 mM potassium ferrocyanide, 1 mg/mL X-gal and 20 mM Tris pH 7.5 in embryo wash buffer). PCR using genomic DNA extracted from embryonic sacs digested with DirectPCR Lysis Reagent (Viagen, 301-C) containing Proteinase K (final concentration 6 U/mL) was used to confirm integration at the H11 locus and test for presence of tandem insertions[16]. Only embryos with donor plasmid insertion at H11 were used. Embryos were not sexed. The stained transgenic embryos were washed three times in PBS and imaged from both sides using a Leica MZ16 microscope and Leica DFC420 digital camera.

### Statistics and reproducibility.

Samples sizes were based on standards in the field. We obtained a minimum of two transgenic embryos with reporter construct integrated at the safe harbor H11 locus and a consistent effect when compared to the wild-type pattern. Randomization was not relevant to this study - each perturbation (mutation) was specific to a given enhancer. No group allocation was performed, and therefore no blinding.

### Reporting summary

Further information on research design is available in the Nature Portfolio Reporting Summary linked to this article.

## Data availability

All raw and processed data relating to the MPRA experiment can be found on the ENCODE portal under ENCSR865OZI (design reference), ENCSR257CZP (association library), ENCSR517VUU (DNA library) and ENCSR548AQS (RNA library). Data related to transgenic assays can be found at VISTA Enhancer Browser (e). Source Data can be found at Gitlab [https://gitlab.com/lotard/mpravista/-/tree/e3de970fc23630e31cd3e4d41ca549781ca4bea8/] or Zenodo (https://doi.org/10.5281/zenodo.15353561).

## Code availability

Code and Source Data for recreating figures in this article can be found at Gitlab [https://gitlab.com/lotard/mpravista/-/tree/e3de970fc23630e31cd3e4d41ca549781ca4bea8/] or Zenodo (https://doi.org/10.5281/zenodo.15353561).

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

## Acknowledgements

This work was supported in part by the National Human Genome Research Institute (NHGRI) grant numbers UM1HG009408 (NA), UM1HG011966 (NA, M.Ki.), R01HG003988 (L.A.P.), the National Institute of Mental Health (NIMH) grant numbers U01MH116438 (NA), R01MH109907 (NA) and National Institute of Child Health and Human Development (NICHD) grant number R01HD114353 (L.A.P). Part of the research was conducted at the E.O. Lawrence Berkeley National Laboratory and performed under U.S. Department of Energy Contract DE-AC02-05CH11231, University of California.

## Author contributions

M.Ko. conceptualized and designed the experiments and performed the analyses. D.L.C. conducted all MPRA-related experiments. M.Ko., D.L.C. and N.A. wrote the manuscript. M.Ko., J.A.A., I.P-F., C.N., M.Ka., R.D.H, K.v.M., S.B., P.G., E.B. conducted transgenic mouse assay experiments. P.K. and M.S. ran BCalm analysis. N.P. ran ABC analysis. I.G.S. ran TFBS analysis. S.S. provided computational resources for ABC analysis. M.Ki. supervised P.K. and M.S. N.A. and L.A.P. provided funding for the study. N.A. and L.A.P. supervised the study.

## Competing interests

N.A. is a cofounder and on the scientific advisory board of Regel Therapeutics. N.A. received funding from BioMarin Pharmaceutical Incorporate. Remaining authors declare no competing interests.
