## [Transparent Peer Review file · Nature Communications]

Massively parallel reporter assays and mouse transgenic assays provide correlated and complementary information about neuronal enhancer activity

Corresponding Author: Professor Nadav Ahituv

Version 0:

Reviewer comments:

Reviewer #1

(Remarks to the Author)

This manuscript reports a large-scale, robust comparison of the results of two widely examined assays for regulatory activity of genomic elements. Massively parallel reporter assays (MPRAs) test the impact of large numbers of relatively short DNA segments on reporter constructs in a specific cell line in a high throughput manner, whereas transgenic mouse reporter assays test the impact of larger DNA segments on reporter gene expression in a wide range of tissues in a mouse embryo, but with a much lower throughput. While it is clear that the two approaches can reveal different types of information, it has not been clear from previous studies to what extent the two assays provided similar or differing results, especially about whether predicted regulatory elements were active in the different assays and whether the assays revealed the impact of phenotype associated, noncoding genetic variants. The experiments reported in this manuscript address these questions directly. The results show that the results of the two assays are strongly correlated and they provide complementary information. Also, the two assays can be used together, leveraging the complementary strengths of each approach. One specific example is to use the high throughput screening of synthetic variants of enhancers to find specific variants (showing a significant difference in MPRA activity) to subsequently examine deeply by transgenic assays, which can reveal loss and gain of function in specific anatomical regions or organs. The experiments are designed well, the results are clearly presented, and the manuscript is lucid and succinct. The results reported here will be of strong interest to investigators studying gene regulation and neurodevelopmental and psychiatric disorders.

The manuscript could be improved by addressing the following points.

(1) It may be helpful to add the fact that the results of the two assays are highly correlated to the title, maybe something like "Massively parallel reporter assays and mouse transgenic assays provide correlated and complementary information about neuronal enhancer activity". While the transgenic reporter assay rightly has been considered one of the more biologically relevant of the common experimental assays for regulatory function, there have been abiding concerns about the relevance of many other assays, including MPRAs. Thus, the fact that a strong correlation was observed should boost confidence in the MPRA results. If that correlation is pointed out in this context, it would be prudent to include the fact that the lentiMPRA employed here assays integrated reporter constructs, and that the correlation may not be as strong for non-integrated MPRAs.

(2) The manuscript should clarify the scope of the MPRA by specifying how many candidate enhancers and/or peaks of chromatin accessibility are represented in the final set of tiles tested. The MPRA elements are 270bp sequences for each tile, and each enhancer or chromatin accessibility peak has multiple tiles across it. The Methods lists several datasets of chromatin accessibility and other data and how many peaks were chosen from them to include in the selection scheme, but it is not clear how many of those peaks are represented in the set of selected tiles.

(3) Lines 161-164: It is not clear which tiles or elements are "non-overlapping" for genomic annotations. It seems like all the categories in Supplementary Figure 2a and Supplementary Table 1 are based on genomic annotations or activities.

(4) Lines 216-221: The panel labels mentioned in the text are off by one letter, apparently: Figure 2b should be Figure 2c, etc.

(5) Line 402: "rDHS" should be explained: is it a "representative DHS"?

To promote transparency, I sign this report:
Ross Hardison

Reviewer #2

(Remarks to the Author)

Many genetic variants have been reported to be associated with psychiatric traits. The authors sought to shed light on function by combining data from MPRA with in vivo transgenic mouse assays. This has yielded a catalog of functional neuronal enhancers and variant effects. Although an important study, there are some concerns:

MAJOR COMMENTS

-Address whether the di-nucleotide scramble sequences were checked for known regulatory motifs. Use of scrambled sequence with known motifs may negatively impact the MPRA results when used as the comparison group and limiting/overestimating individual mutagenized sites. (maintaining di-nucleotide frequency for scramble is good)

-RNA counts/DNA counts indicate sum. Is this correct? If so, please show that there were no extreme outliers for barcode counts representing an MPRA tile. i.e. one MPRA tile has one barcode counted 500 times while 14 barcodes for that MPRA tile are each counted around 25 times.

-Justify the use of MPRA median values as opposed to mean. Again, what is the distribution of the MPRA element counts, and what does the mean look like in comparison to the median?

-line 451 indicates an average of 100 barcodes per sequence, please provide a visual supplemental assessment of total barcode counts per sequence.

-Different genome builds were used when looking at MPRA tiles (hg38) vs ATAC-seq (hg19) data that was used for ABC score for candidate genes. Results shown starting line 277, methods indicated in line 411 and line 551. When using the ABC score for choosing candidates related to MPRA results that were based on hg38, how does the different builds impact the results (specifically when working with the ABC score). Is this limiting potential targets? And by extension, limiting the total potential numbers of MPRA sites that were assessed?

- 280-281 indicates how the transgenic follow-up was chosen, which was predominantly focused on ABC scoring within the defined cell line. Those were then further reduced based on MPRA activity scoring. Out of the 777 original MPRA signals, you only retained 7, one of which was indicated as not meeting the defined threshold of significance. Why was this retained for further evaluation? ("hs2792.1" element).

MINOR COMMENTS

-Line 48 missing citation for de novo statement.

-line 245, Negatively/inversely correlated?

-Line 272, please indicate how to interpret the use of "unit" when considering the impact on expression. Is this supposed to represent the log₂FC?

-Line 355, "Their remaining shortcomings can be overcome by combining the techniques." Redefine "their" when referencing MPRA/transgenic. Also, what shortcomings to MPRA and transgenic techniques are being referenced in this?

-Line 372 citations

-Provide justification for comparing the 'active' MPRA variants in differentiated human excitatory neurons derived from an isogenic WTC11-Ngn2 iPSC line with ATAC/DNAse datasets representing "diverse tissues and cell types" line 212.

-Line 300 "Correlation" appears inappropriate as there is no statistical analysis to justify this term usage when referring to MPRA and transgenic assay. Please re-phrase

Reviewer #3

(Remarks to the Author)

Review of "Massively parallel reporter assays and mouse transgenic assays provide complementary information about neuronal enhancer activity" by Kosicki et al for Nature Communications.

The authors present a large lentiMPRA dataset testing enhancer candidates selected based on prior transgenic mouse assays and neuronal ATAC datasets, including sequence variants (synthetic and from GWAS). They test seven variants in

transgenic mouse assays and, based on this test and a comparison between the MPRA results and the VISTA database, report that they “found a strong and specific correlation between MPRA and mouse neuronal enhancer activity” and “pleiotropic variant effects” (both citations from the abstract).

The manuscript addresses two important questions, namely the concordance of MPRA and in vivo transgenic mouse assays and the gene-regulatory impact of GWAS hits and mutations in cis-regulatory elements more generally. While I find the effort commendable, the MPRA dataset seems to be noisy and presumably not statistically significant – a proper statistical assessment is lacking (see below). Moreover, some analyses might be confounded by prior candidate selection and the comparison between MPRA and in vivo data appears to be weak (Fig. 3b) or anecdotal (Table 1 & Fig. 4). Unfortunately, this means that the authors’ main claims, including the two cited above, are not supported by the data.

Major concerns

1. Inappropriate statistical analysis and noisy MPRA data

The authors test 81,952 elements of which 76,414 pass their QC (page 6 and methods) yet evaluate the statistical significance by nominal p-values from a t-test with a lenient cutoff of 0.05. The use of nominal p-values without multiple-testing correction when performing >70,000 tests is inappropriate and suggests that the majority of the “activators” and “repressors” (page 6) are false positives, also because the overall correlation between replicates is rather poor for MPRA (0.58-0.59). Multiple-testing corrected FDRs are not available for this dataset, but later in the manuscript (page 11), the authors state that none of the 3 GWAS hits (of 177 tested) remain significant after multiple testing adjustment. I am concerned that the authors use a sub-standard and inappropriate statistical analysis and that most of the resulting hits might be false positives.

2. Potentially confounded analyses

I’m afraid that the results presented in Fig. 2 (page 8 -9) might be confounded by the prior selection of candidate elements to overlap VISTA enhancers or ATAC regions. As these regions don’t evenly overlap the different genomic regions (Fig. 2a) and have distinct motif content from the average genome (Fig. 2b) some of the results might stem from candidate pre-selection rather than MPRA results. This is best illustrated by the motif analysis, which reveals that enriched motifs can only be found when comparing to genomic background but not to tiles with background-level activity (page 8). The authors briefly acknowledge this problem on page 9, but don’t control for it.

I also wonder if the inclusion of conserved tiles as a covariate in the model (Fig. 3b, c; page 9) is required, what it does and if it does not even create signals artificially. MPRA and mouse transgenic assays are blind to sequence conservation. I also note that the data in Fig. 3b indicate a large overlap in MPRA activity between elements that are active or inactive in VISTA – is there a significant difference and what is the effect size of this difference?

3. Anecdotal follow up experiments

The follow up analysis with transgenic mouse assays is anecdotal at best. While the abstract makes a quite definite and general claim (“strong and specific correlation between MPRA and mouse neuronal enhancer activity”), only 7 elements were tested. Moreover, the selection of these elements are highly questionable: while the abstract speaks of “variants with a high impact on MPRA activity were further tested in mice”, this is actually not true. Instead, the authors “prioritized variants with links to important neuronal genes” (page 11), which even included one example with statistically insignificant variant effect. It remains unclear how this selection – rather than the MPRA results – impact the results.

Moreover, as for the overall analysis, nominal p-values and a lenient cutoff (0.05) were used even though 23,266 non-GWAS variants were tested (and 777 found) – no multiple testing adjustment is reported.

I also note that the reported in vivo results (Fig. 4b) are difficult to see and assess.

4. Unsupported claims and potentially misleading statements

As partly indicated above, the manuscript contains many unsupported claims and potentially misleading statements. Given that only seven variants were tested in transgenic mouse assays and that these were selected based on links to neuronally relevant genes, statements like “strong and specific correlation between MPRA and mouse neuronal enhancer activity”, “variants with a high impact on MPRA activity were further tested in mice” (both abstract) or “most synthetic variants nominated by MPRA as having a large impact also affected the transgenic assay activity in the expected manner” (discussion, page 14) are incorrect and misleading.

Minor comments

When following the main text, the statement “only tiles with at least 15 barcodes detected in all three replicates were retained” (page 6) is unclear. At this point in the text, it is not clear that the authors’ lentiMPRA includes barcodes (maybe add this to the method description just above on lines 126-127). Also, neither here nor in the methods (page 19) do the authors explain if this cutoff is applied to the DNA (library representation) or RNA (element activity) or both. Obviously, only the first, library representation, should have this cutoff.

For motif enrichment analyses to work via three criteria (fraction of motifs in activator tiles, increase by >50%, and FDR-adjusted p-value <0.01) is unusual (though not incorrect!). Maybe the authors can briefly explain this choice.

On page 14 (discussion), the authors speculate that a potential reason for not observing a significant effect on MPRA activity for GWAS variants could be that they tested only a small number of variants. While it might be true that they missed variants with true effects by their selection, the study already suffers from lack of statistical significance due to multiple testing.

Collegial recommendation

Independent of this peer review, I'd recommend that the authors rigorously assess their lentiMPRA method, which appears to be the source of the poor reproducibility and lack of statistical significance. One potential problem that the authors did not test here or in ref 14 is that lentiviruses preferentially integrate into transcribed regions via their LEDGF/p75 PWWP domains (e.g. Gijsbers et al., JBC 2011). This might confound both CRE activity and barcode quantification.

Reviewer #4

(Remarks to the Author)

In this study Kosicki et al. conduct a lenti-viral-based massively parallel reporter assay in iPSC-derived neurons. Over 50,000 sequences were assayed, spanning several different sequence groupings, such as putative brain, heart, and kidney enhancers. In addition to wildtype sequences, synthetic and disease associated variants were also tested. Neuronal enhancers had elevated activity in the MPRA compared to other sequence groups, and within sequences derived from VISTA elements tested in mouse transgenic assays, the MPRA activity correlated with transgenic assay activity. In addition, the authors find that variants that alter MPRA activity also affected neuronal enhancer activity in mouse transgenic assays. Collectively, these results support the idea that for at least certain cell types, MPRA's conducted in vitro can be used to functionally filter large numbers of sequences, to identify and analyze tissue specific enhancers. This work is significant as it addresses a currently understudied area - the correlation between MPRA data derived from short sequences tested in vitro and mouse transgenic assays, which remain the gold standard due to their ability to generate in vivo spatio-temporal data from full length enhancers. Overall, the study provides valuable context for the interpretation of MPRA's, and supports their value as a high-throughput tool. There are, however, several areas that need clarification.

Line 280 "We prioritized variants with links to important neuronal genes ..." This appears to have led to selection of elements for the transgenic assay that had low activity in the MPRA. For example, the activity of the reference element for hs268.1 was lower than the average negative control element, and the variant was expected to have even lower activity. Indeed, of the seven variants tested, the three with a high baseline level of activity (>3) all had at least some loss of expression, while only one of the four with low baseline activity (<2) had a loss of expression. Given that figure 3 showed that model validation rate increased with increasing MPRA activity, and a large part of the rationale for the study is to demonstrate the usefulness of using MPRA as a variant filter for transgenic mouse assays, the decision to select these low baseline activity elements for the transgenic assay is puzzling. Additional elements with higher MPRA baseline activity should be tested.

No follow up experiments were done to validate findings of the MPRA. For example, 13 motifs were observed to be enriched in high activity elements; a follow up MPRA experiment featuring ablation of these motifs, or synthetic elements with different combinations of the enriched motifs would reveal insights into neural transcriptional grammar.

Generation of new transgenics was limited to variants that caused loss of function in the MPRA. The impact of the study would be enriched by showcasing the ability of MPRA's to find sequences that would be excluded from transgenic creation if solely non-functional filtering criteria were used (e.g. excluded if the region didn't have a classical enhancer chromatin signature). I would not require it, but making some transgenics from novel sequences that showed robust MPRA activity would significantly enhance the study.

Housekeeping promoters were highly active in the MPRA, and it's suggested that they can be used as universal positive controls in other MPRA's. It's important to note that since they are cloned upstream of the minimal promoter they will function as normal promoter elements, directly driving expression of the reporter gene (this would not be the case in a different MPRA format such as STARRseq). Thus, their high activity is probably not due to enhancer activity. This could be tested by looking at the orientation that promoters were inserted, and comparing the activity of promoters cloned in the forward orientation to promoters cloned in the reverse orientation, which presumably could not directly initiate transcription of the reporter gene.

Supplementary tables 2-4 were not included in the reviewer file.

Line 190 – It would be nice to include images of the motifs for the 13 motifs found to be enriched, as done in Fig4c,d. Adding de novo motif discovery to the analysis would also be interesting.

In Figure 4a the variant tested by transgenic assay is colored in blue, which is difficult to see. Consider a more distinct color.

The epigenetic signal is described as coming from "WTC11 datasets" on line 211, but then described as "differentiated WTC11 cells" on line 221. I guess they are both referring to differentiated WTC11 cells, but labelling this group as only WTC11 on the graph in Fig. 2d is confusing. I would suggest changing it to "WTC11 Neurons" or something similar.

The panel references in the text for Figure 2 appear to be incorrect. For example, it looks like line 216 Fig 2b should actually be referencing Fig. 2d. "Fig. 2d left" on line 221 should be "Fig. 2d right".

Reviewer #5

(Remarks to the Author)

Version 1:

Reviewer comments:

Reviewer #1

(Remarks to the Author)

The manuscript has been revised to address all the concerns I raised in the initial review. Furthermore, improved statistical analyses were employed that handled data variability more thoroughly and implemented a multiple-testing correction, which reduced the number of tiles designated as activators and repressors. This improved analysis, along with several additional experiments and analyses, strengthened this manuscript.

Signing this review,
Ross Hardison

Reviewer #2

(Remarks to the Author)

Major and minor comments provided to authors have been addressed or, at minimum, clarified within the manuscript. This has enhanced readability. Figures added to supplement has also been included, adding further understanding of the underlying methodologies utilized within this project.

Reviewer #3

(Remarks to the Author)

The authors have provided answers to our questions and concerns. We have no further comments.

Reviewer #4

(Remarks to the Author)

This revision features improvements in several aspects of the analysis, including improved barcode handling, resulting in less variability between MPRA replicates, a more robust motif analysis, and additional comparisons with pre-existing MPRA datasets. Additional transgenic data was also added, and several areas in the original manuscript that were confusing have received clarifying text and are now significantly improved. I still think that basing candidate selection for the variant transgenic assay on evidence of neuronal activity in transgenic assays and links to neuronal genes, rather than focusing on the MPRA activity, is a missed opportunity. However, I am sympathetic to the explanation given and find that the revisions to this section of the manuscript are a substantial improvement. I believe my critiques have been sufficiently addressed, and have only one minor comment regarding line 288: "we selected 6 elements with an average predicted likelihood of in vivo neuronal activity of 86% and tested them using transgenic mouse assays." It would be helpful to add the MPRA activity of each tested element to Supplemental Figure 4.

Reviewer #5

(Remarks to the Author)

We thank the reviewers for their great comments, which have significantly improved our revised manuscript. Below we provide in blue font our response to these comments.

In summary, we have re-analyzed our MPRA data by filtering out barcodes with high inter-replicate variability and using the BCalm package for barcode-level statistical testing, which substantially improved inter-replicate correlation coefficients (from on average 0.59 to 0.77) and allowed us to detect many elements and variants with statistically significant activity after multiple-testing corrections. Furthermore, we have validated our MPRA-based modeling approach for predicting elements with *in vivo* activity both experimentally (adding 6 transgenic experiments) and using independent, published MPRA datasets.

Reviewer #1

This manuscript reports a large-scale, robust comparison of the results of two widely examined assays for regulatory activity of genomic elements. Massively parallel reporter assays (MPRAs) test the impact of large numbers of relatively short DNA segments on reporter constructs in a specific cell line in a high throughput manner, whereas transgenic mouse reporter assays test the impact of larger DNA segments on reporter gene expression in a wide range of tissues in a mouse embryo, but with a much lower throughput. While it is clear that the two approaches can reveal different types of information, it has not been clear from previous studies to what extent the two assays provided similar or differing results, especially about whether predicted regulatory elements were active in the different assays and whether the assays revealed the impact of phenotype associated, noncoding genetic variants. The experiments reported in this manuscript address these questions directly. The results show that the results of the two assays are strongly correlated and they provide complementary information. Also, the two assays can be used together, leveraging the complementary strengths of each approach. One specific example is to use the high throughput screening of synthetic variants of enhancers to find specific variants (showing a significant difference in MPRA activity) to subsequently examine deeply by transgenic assays, which can reveal loss and gain of function in specific anatomical regions or organs. The experiments are designed well, the results are clearly presented, and the manuscript is lucid and succinct. The results reported here will be of strong interest to investigators studying gene regulation and neurodevelopmental and psychiatric disorders.

Thank you for your comments and suggestions for improvements, this is much appreciated!

The manuscript could be improved by addressing the following points.

(1) It may be helpful to add the fact that the results of the two assays are highly correlated to the title, maybe something like "Massively parallel reporter assays and mouse transgenic assays provide correlated and complementary information about neuronal enhancer activity". While the transgenic reporter assay rightly has been considered one of the more biologically relevant of the common experimental assays for regulatory function, there have been abiding concerns about the relevance of many other assays, including MPRAs. Thus, the fact that a strong correlation was observed should boost confidence in the MPRA results. If that correlation is

pointed out in this context, it would be prudent to include the fact that the lentiMPRA employed here assays integrated reporter constructs, and that the correlation may not be as strong for non-integrated MPRAs.

We revised the title to the proposed one: "Massively parallel reporter assays and mouse transgenic assays provide **correlated and** complementary information about neuronal enhancer activity". Regarding lenti-based MPRA versus episomal, there are definitely differences between these experiments as shown by our labs in the past (PMIDs 27831498 and 33046894) but overall they correlate well (see for example Figure 2b from PMID 33046894 where episomal (pGL4) and lentiMPRA (5'/5' WT) have a 0.87 Pearson correlation for the same library). We have added the following text to the Discussion to mention this:

"We note that we only used lentiMPRA here and other MPRAs technologies could potentially lead to different results though they are usually well correlated³⁸."

(2) The manuscript should clarify the scope of the MPRA by specifying how many candidate enhancers and/or peaks of chromatin accessibility are represented in the final set of tiles tested. The MPRA elements are 270bp sequences for each tile, and each enhancer or chromatin accessibility peak has multiple tiles across it. The Methods lists several datasets of chromatin accessibility and other data and how many peaks were chosen from them to include in the selection scheme, but it is not clear how many of those peaks are represented in the set of selected tiles.

We added these numbers to the manuscript as Supplementary Table 6, thank you for the suggestion! Since these peaks are not mutually exclusive (a peak in e11.5 forebrain is likely to also be in e11.5 midbrain etc), we also added descriptive statistics to give the reader an idea of the coverage:

"Together, the elements covered 11.7 Mbp of genomic sequence in 24,000 non-overlapping regions of 270 bp (tile size) to 6531 bp (mean 488 bp)".

(3) Lines 161-164: It is not clear which tiles or elements are "non-overlapping" for genomic annotations. It seems like all the categories in Supplementary Figure 2a and Supplementary Table 1 are based on genomic annotations or activities.

Since this analysis was confusing, we have decided to remove it from the updated manuscript.

(4) Lines 216-221: The panel labels mentioned in the text are off by one letter, apparently: Figure 2b should be Figure 2c, etc.

Thanks for catching this! We corrected this.

(5) Line 402: "rDHS" should be explained: is it a "representative DHS"?

Yes, now expanded. We also corrected the citation for that dataset, it's Moore 2020, not Meuleman 2020.

Reviewer #2

Many genetic variants have been reported to be associated with psychiatric traits. The authors sought to shed light on function by combining data from MPRA with in vivo transgenic mouse assays. This has yielded a catalog of functional neuronal enhancers and variant effects. Although an important study, there are some concerns:

MAJOR COMMENTS

-Address whether the di-nucleotide scramble sequences were checked for known regulatory motifs. Use of scrambled sequence with known motifs may negatively impact the MPRA results when used as the comparison group and limiting/overestimating individual mutagenized sites. (maintaining di-nucleotide frequency for scramble is good).

We thank the reviewer for this comment. We did not specifically analyze the scrambled sequences for regulatory motifs. Such an analysis would not tell us if these sequences are active / repressive, so it would not inform us as to whether scramble negative mean/variance are correctly estimated. We did not use scramble negatives as background control for our motif analysis, so that should not impact estimates of *individual mutagenized sites*. As an alternative approach, we now included a comparison between scrambled negatives and genomic regions from which scrambled negatives were derived with the following text:

"Negative control reference tiles, which were selected from non-conserved parts of elements negative in transgenic assay and with no epigenomic signal in neural datasets, had a similar activity to their scrambled counterparts (Supplementary Figure 1b,c). This both validated their selection strategy and showed that scrambling did not systematically make elements active or repressive."

-RNA counts/DNA counts indicate sum. Is this correct? If so, please show that there were no extreme outliers for barcode counts representing an MPRA tile. i.e. one MPRA tile has one barcode counted 500 times while 14 barcodes for that MPRA tile are each counted around 25 times.

We have investigated the relationship between mean and median number of UMIs per barcode (see Figure below). Strong deviations from the diagonal would indicate presence of outliers. None of the elements in any of the replicates or modalities seems to be far outside the diagonal. We followed up the three most extreme elements (highlighted in green, blue and orange) and found none of them had even nominally statistically significant activity (smallest p-value was around 0.2). We conclude detection of individual barcodes with unusual levels of UMIs is unlikely to significantly influence the results of this MPRA.

-Justify the use of MPRA median values as opposed to mean. Again, what is the distribution of the MPRA element counts, and what does the mean look like in comparison to the median?

The consistent use of median in this manuscript is motivated by the desire to provide an easily interpretable (50% above, 50% below), non-parametric estimate of activities of various groups of elements, and to make it consistent with graphical boxplot representations. The distribution of MPRA activities is provided as inset in Fig 3a. The mean and the median activity of the library is nearly the same, with mean = 0.22 and median = 0.21 (for reference elements only it's 0.14 and 0.15, respectively - the second number is crucial to interpretation of Fig 2a for example). The higher median of the whole library compared to the reference is due to selection of reference tiles for synthetic variant mutagenesis (which take up the majority of non-reference elements) based on high likelihood of activity.

-line 451 indicates an average of 100 barcodes per sequence, please provide a visual supplemental assessment of total barcode counts per sequence.

Added as Supplementary Fig 1a.

-Different genome builds were used when looking at MPRA tiles (hg38) vs ATAC-seq (hg19) data that was used for ABC score for candidate genes. Results shown starting line 277, methods indicated in line 411 and line 551. When using the ABC score for choosing candidates related to MPRA results that were based on hg38, how does the different builds impact the results (specifically when working with the ABC score). Is this limiting potential targets? And by extension, limiting the total potential numbers of MPRA sites that were assessed?

ABC scores were not used to select which tiles to introduce synthetic variants into (L411). We also did not have these available at the time of the design. They did partially inform the choice of variant to test in the transgenic assay, but prior evidence of neural activity of the reference element in transgenic assay and impact of the variant in MPRA were weighed more. We clarify this aspect of variant selection in the revised manuscript:

"We selected five of these variants for follow up in the transgenic assay, based on prior evidence of neuronal activity in transgenic assays and, to a lesser degree, links to important neuronal genes predicted using the ABC model (e.g. QKI, PRKN, COA7 and MEF2C; Table 1, Figure 4a)"

Furthermore, we clarify in the methods that we lifted over the ABC results to hg38:

"Promoter enhancer contacts from ABC score were saved as bedpe files and coordinates were mapped from hg19 to hg38 using liftOverBedpe (<https://github.com/dphansti/liftOverBedpe>). The genomic coverage of hg38 compared to hg19 is only slightly increased (95% vs ~92.5%).

- 280-281 indicates how the transgenic follow-up was chosen, which was predominantly focused on ABC scoring within the defined cell line. Those were then further reduced based on MPRA activity scoring. Out of the 777 original MPRA signals, you only retained 7, one of which was indicated as not meeting the defined threshold of significance. Why was this retained for further evaluation? ("hs2792.1" element).

Our initial selection of elements was based on an early analysis of the MPRA which was later updated, leading to dropout of hs2792.1. Based on this comment, we decided to remove it from the revised manuscript to avoid confusion.

MINOR COMMENTS

-Line 48 missing citation for de novo statement.

Added Goes 2021 (PMID:31776463).

-line 245, Negatively/inversely correlated?

Changed to 'negatively'.

-Line 272, please indicate how to interpret the use of “unit” when considering the impact on expression. Is this supposed to represent the log₂FC?

Thank you for pointing out this lack of clarity - everything in the paper is expressed in "scrambled negative standard deviations" (as in z-score vs scrambled negatives distribution), so we added this information to the two spots where "unit" was used (L272 and L283).

-Line 355, “Their remaining shortcomings can be overcome by combining the techniques.”
Redefine “their” when referencing MPRA/transgenic. Also, what shortcomings to MRPA and transgenic techniques are being referenced in this?

To streamline this section, we replaced this sentence with "However, both methods still suffer from significant shortcomings.", after which we list the shortcomings, as before.

-Line 372 citations

Added Levo 2020, "Transcriptional coupling of distant regulatory genes in living embryos".

-Provide justification for comparing the ‘active’ MPRA variants in differentiated human excitatory neurons derived from an isogenic WTC11-Ngn2 iPSC line with ATAC/DNase datasets representing “diverse tissues and cell types” line 212.

We have rephrased this sentence to indicate the intention to validate MPRA signal as specific: "To investigate whether the activity signal in our MPRA is biologically specific, we compared our MPRA results to epigenomic signal from diverse tissues and cell types from 12 embryonic, fetal and WTC11 datasets, encompassing 740 Dnase hypersensitive sites (DHS), ATAC and single-cell ATAC samples."

-Line 300 “Correlation” appears inappropriate as there is no statistical analysis to justify this term usage when referring to MPRA and transgenic assay. Please re-phrase

Changed to "correspondence".

Reviewer #3

Review of “Massively parallel reporter assays and mouse transgenic assays provide complementary information about neuronal enhancer activity” by Kosicki et al for Nature Communications.

The authors present a large lentiMPRA dataset testing enhancer candidates selected based on prior transgenic mouse assays and neuronal ATAC datasets, including sequence variants (synthetic and from GWAS). They test seven variants in transgenic mouse assays and, based on this test and a comparison between the MPRA results and the VISTA database, report that they “found a strong and specific correlation between MPRA and mouse neuronal enhancer activity” and “pleiotropic variant effects” (both citations from the abstract).

The manuscript addresses two important questions, namely the concordance of MPRA and in vivo transgenic mouse assays and the gene-regulatory impact of GWAS hits and mutations in cis-regulatory elements more generally. While I find the effort commendable, the MPRA dataset seems to be noisy and presumably not statistically significant – a proper statistical assessment is lacking (see below). Moreover, some analyses might be confounded by prior candidate selection and the comparison between MPRA and in vivo data appears to be weak (Fig. 3b) or anecdotal (Table 1 & Fig. 4). Unfortunately, this means that the authors’ main claims, including the two cited above, are not supported by the data.

Thank you for your critical comments. We have now re-analyzed our MPRA data by filtering out barcodes with high inter-replicate variability and using the BCalm package (<https://doi.org/10.1186/s12859-025-06065-9>) for statistical analysis on barcode-level, which substantially improved inter-replicate correlation coefficients (from on average 0.59 to 0.77) and allowed us to detect many elements and variants with statistically significant activity after multiple-testing corrections. See more details below.

Major concerns

1. Inappropriate statistical analysis and noisy MPRA data

The authors test 81,952 elements of which 76,414 pass their QC (page 6 and methods) yet evaluate the statistical significance by nominal p-values from a t-test with a lenient cutoff of 0.05. The use of nominal p-values without multiple-testing correction when performing >70,000 tests is inappropriate and suggests that the majority of the “activators” and “repressors” (page 6) are false positives, also because the overall correlation between replicates is rather poor for MPRA (0.58-0.59). Multiple-testing corrected FDRs are not available for this dataset, but later in the manuscript (page 11), the authors state that none of the 3 GWAS hits (of 177 tested) remain significant after multiple testing adjustment. I am concerned that the authors use a sub-standard and inappropriate statistical analysis and that most of the resulting hits might be false positives.

As mentioned above, our new analytical approach using BCalm package allowed us to deal with the noise in our dataset, uncovering over a thousand tiles with activity significantly different from scrambled negatives and 800 variants with significant impact on element activity (after FDR correction). Furthermore, our analyses of transcription factor motif content, enrichment for epigenomic signal from neuronal tissues (Fig 2) and correlation with *in vivo* transgenic assay activity in neuronal tissues (Fig 3) further demonstrate that our measurements captured specific signals.

2. Potentially confounded analyses

I'm afraid that the results presented in Fig. 2 (page 8 -9) might be confounded by the prior selection of candidate elements to overlap VISTA enhancers or ATAC regions.

We thank the reviewer for this comment. All of the analyses presented in Fig 2 have internal controls. First, we use dinucleotide scrambled elements derived from regions with low prior likelihood of activity as controls in most of our analyses. Second, we conducted HOMER transcription factor motif enrichment analyses using both GC-matched genomic regions and regions with scramble negative-like activity in our MPRA, finding similar sets of motifs. Third, we find much stronger enrichment for WTC11 neuronal epigenomic signal in tiles with high MPRA activity, not uniform enrichment across the entire activity spectrum. All of these suggest we are controlling for design appropriately.

Furthermore, a design consisting of random genomic regions would result in a tiny fraction of the library having any activity. Such a design would severely limit the power to answer biologically relevant questions. This is why most MPRA designs (if not all), including this one, aim to enrich for active elements.

As these regions don't evenly overlap the different genomic regions (Fig. 2a) and have distinct motif content from the average genome (Fig. 2b) some of the results might stem from candidate pre-selection rather than MPRA results. This is best illustrated by the motif analysis, which reveals that enriched motifs can only be found when comparing to genomic background but not to tiles with background-level activity (page 8). The authors briefly acknowledge this problem on page 9, but don't control for it.

This is an astute observation. It is correct that for repressor elements we find enriched TFBSs compared to genomic background, but not to elements with scramble negative levels of activity within the library, which could potentially indicate lack of specific MPRA signal. To address this and following also Reviewer 4's suggestion we also performed *de novo* motif search and added these sentences that expand on these analyses: "Repressor tiles were enriched for Nkx6.1 and four motifs from SOX family, but only in analysis using genomic background. This may imply lack of specific repressive signal in our library, limited power to detect such signal or relative dearth of known repressive motifs in the the HOMER dataset. The latter might be consistent with *de novo* motif analysis conducted against tiles with background activity, which revealed similar motifs for activator tiles (RFX7/Rfxdc2, X-box, SP5, match scores 0.82-0.9), but novel

repressor motifs with tentative matches to ZIC2, KLF15, and SP1 among others (match scores of 0.7-0.75; Supplementary Table 3).".

On the other hand, we observed that it is incorrect that "enriched motifs can only be found when comparing to genomic background" for activator tiles. In fact, comparison of activators to regions with scramble negative levels of activity yields strong enrichments of relevant TFBSs. This within-library controlled analysis specifically demonstrates that measured MPRA activity is informative and not simply a consequence of biased region selection.

I also wonder if the inclusion of conserved tiles as a covariate in the model (Fig. 3b, c; page 9) is required, what it does and if it does not even create signals artificially. MPRA and mouse transgenic assays are blind to sequence conservation.

Great comment and we think one that depends on the definition of "blind". Evolutionary conservation is one of the strongest correlates of functionality and is routinely used as means of selecting elements with activity in transgenic assays and selecting variants likely to affect biology (e.g. Pennacchio 2006, PMID:17086198).

We attempted to regress or marginalize conservation, because we were concerned that our experimental design would unduly bias the modeling. This design bias is visualized in Figure 3a. We also added the following text to the Results section to explain it more clearly:

"In our design, we have included negative control tiles derived from non-conserved parts of negative VISTA elements that did not overlap epigenomic signal from any of the neural datasets. Conversely, we aimed to capture as many conserved parts of positive VISTA elements as possible. To account for that design bias, we included a fraction of conserved sequences covered by tiles as a covariate in the model (Figure 3b,c).".

We have now added two analyses as evidence that including conservation in the model is necessary to control for experimental design. First, we showed that an uncontrolled analysis found a strong, significant anticorrelation between MPRA activity and negative VISTA elements (Supplementary Figure 3d), unlike the controlled analysis. Second, both conservation-controlled and uncontrolled analyses using an independent, published MPRA conducted in primary human fetal cortical cells (PMID: 38781390) do not show any enrichment for negative VISTA elements, like the controlled analysis of our MPRA. We describe these additional analyses as follows:

"In contrast, an uncontrolled model, while showing similar neuronal and heart term enrichments, also showed a strong negative correlation with negative VISTA elements, demonstrating the impact of uneven sampling of conserved regions in positive and negative elements (Supplementary Figure 3d)"

"Similar to our MPRA, forebrain and combined brain terms were positively correlated with MPRA activity [in PMID: 38781390], while heart, heart+somite and facial-mesenchyme were negatively correlated (Supplementary Figure 3e). We note that for this MPRA, a model taking into account

sequence conservation across species was very similar to the model that did not. This likely reflects the fact that conservation signal was not used to select regions for testing in this MPRA (Supplementary Figure 3e)."

I also note that the data in Fig. 3b indicate a large overlap in MPRA activity between elements that are active or inactive in VISTA – is there a significant difference and what is the effect size of this difference?

The mean activity of tiles associated with neural-positive VISTA elements in Fig. 3b is 0.97 and of the remaining tiles is 0.39 (difference = 0.58, t-test values: $t = 16.5$, $df = 1374$, $p\text{-value} < 2.2e-16$, ie smaller than the default floating point precision; same p-value for non-parametric Mann-Whitney U test). This statistic is likely heavily confounded by the bias in conservation coverage (see our response to the previous comment).

3. Anecdotal follow up experiments

The follow up analysis with transgenic mouse assays is anecdotal at best. While the abstract makes a quite definite and general claim ("strong and specific correlation between MPRA and mouse neuronal enhancer activity"), only 7 elements were tested.

We apologize if this wasn't clear in the article. We now modified the abstract to avoid confusion between the general linear model analysis (that shows a strong correlation) and variant testing. We also removed the non-significant MPRA variants from variant testing section. The relevant part of the abstract now reads:

"We found a strong and specific correlation between MPRA and mouse neuronal enhancer activity. Four out of five tested variants with significant MPRA effects affected neuronal enhancer activity in mouse embryos."

Moreover, the selection of these elements are highly questionable: while the abstract speaks of "variants with a high impact on MPRA activity were further tested in mice", this is actually not true. Instead, the authors "prioritized variants with links to important neuronal genes" (page 11), which even included one example with statistically insignificant variant effect. It remains unclear how this selection – rather than the MPRA results – impact the results.

We apologize that this phrasing was misleading. We removed the non-significant variants from the study, the sentence about the high impact from the abstract and made it clearer that our selection procedure prioritized evidence of transgenic assay activity:

"We selected five of these variants for follow up in the transgenic assay, based on prior evidence of neuronal activity in transgenic assays and, to a lesser degree, links to important neuronal genes predicted using ABC model (e.g. QKI, PRKN, COA7 and MEF2C; Table 1, Figure 4a)"

We also refrained from commenting on how surprising that rate was to us and left it for the readers to judge. Estimating how much exactly using MPRA improves the predictive value is part of ongoing studies of human pathogenic variants that our labs are conducting together.

Moreover, as for the overall analysis, nominal p-values and a lenient cutoff (0.05) were used even though 23,266 non-GWAS variants were tested (and 777 found) – no multiple testing adjustment is reported.

After updating our analysis, "Out of 20,126 successfully tested non-GWAS variants (within 22,710 variant tiles), 751 had a significant effect on regulatory activity (FDR-corrected p-value < 0.05, absolute log2 fold-change > 1)."

I also note that the reported in vivo results (Fig. 4b) are difficult to see and assess.

We increased the size of panel 4b.

4. Unsupported claims and potentially misleading statements

As partly indicated above, the manuscript contains many unsupported claims and potentially misleading statements. Given that only seven variants were tested in transgenic mouse assays and that these were selected based on links to neuronally relevant genes, statements like "strong and specific correlation between MPRA and mouse neuronal enhancer activity", "variants with a high impact on MPRA activity were further tested in mice" (both abstract) or "most synthetic variants nominated by MPRA as having a large impact also affected the transgenic assay activity in the expected manner" (discussion, page 14) are incorrect and misleading.

With the exception of the first statement ("strong and specific"), which is based on correlational analysis, not variant testing, we toned down or made these (and related) statements more precise as follows:

- "variants with a high impact on MPRA activity were further tested in mice" - deleted
- "most synthetic variants nominated by MPRA as having a large impact also affected the transgenic assay activity in the expected manner" - 'Four out of five synthetic variants nominated by MPRA as having a significant impact also affected the transgenic assay activity...'
- abstract: "We found a strong and specific correlation between MPRA and mouse neuronal enhancer activity. Four out of five tested variants with significant MPRA effects affected neuronal enhancer activity in mouse embryos."

Minor comments

When following the main text, the statement "only tiles with at least 15 barcodes detected in all three replicates were retained" (page 6) is unclear.

We rephrased it as "detected in *each of the three replicates*" in main text and Methods.

At this point in the text, it is not clear that the authors' lentiMPRA includes barcodes (maybe add this to the method description just above on lines 126-127).

Good point, now included ("...cloned into a *barcoded* lentiMPRA vector...")

Also, neither here nor in the methods (page 19) do the authors explain if this cutoff is applied to the DNA (library representation) or RNA (element activity) or both. Obviously, only the first, library representation, should have this cutoff.

The MPRAflow pipeline makes an assumption that a barcode observed in RNA likely exists in the reporter in DNA of the cell, and vice versa. We clarified that now in Methods:
"As per MPRAflow pipeline, these [15] barcodes include ones detected in DNA or RNA. In other words, barcodes detected using only one modality were not discarded."

For motif enrichment analyses to work via three criteria (fraction of motifs in activator tiles, increase by >50%, and FDR-adjusted p-value <0.01) is unusual (though not incorrect!). Maybe the authors can briefly explain this choice.

Going only by the customary p-value < 0.05 of the hypergeometric test on thousands of tiles would yield 100s of motifs (hypergeometric tests are conservative), and include statistical enrichments that are less relevant for the big picture of WTC11 neuronal function (e.g. fraction of elements containing a motif going from 2% to 3%). Specifically, such analysis would yield 214 significant motifs, which is 49% of all motifs on which the analysis was run. We included these strict filtering criteria to be able to discuss the results in a more meaningful way, but we also reported all the relevant statistics so as to make it easy for the reader to play around with the cutoffs as they see fit.

On page 14 (discussion), the authors speculate that a potential reason for not observing a significant effect on MPRA activity for GWAS variants could be that they tested only a small number of variants. While it might be true that they missed variants with true effects by their selection, the study already suffers from lack of statistical significance due to multiple testing.

We now detect 7 GWAS variants with significant impact after FDR-correction. We also provide additional discussion on why we might be missing effects for more variants:
"We observed a significant effect on MPRA activity for 7 out of 167 tested psychiatric disorder associated GWAS variants (4.2%), corresponding to 11/117 GWAS loci (9.4%). This is in line with another MPRA carried out by our lab that found 164 psychiatric disorder and eQTL variants out of 17,069 tested (< 1%) to have an effect on MPRA activity⁵⁰. It is unclear how many of the GWAS signals are driven by regulatory elements, but it is possible we missed regulatory variants with lower effect sizes, associated with non-transcriptional phenotypes, like chromatin tethering⁵⁶, or variants having an effect in another cell type or at a different differentiation time point. We note that machine learning models of MPRA data and saturation mutagenesis experiments^{31,50} show that rare variants have a higher effect on MPRA activity compared to common variants."

Collegial recommendation

Independent of this peer review, I'd recommend that the authors rigorously assess their lentiMPRA method, which appears to be the source of the poor reproducibility and lack of statistical significance. One potential problem that the authors did not test here or in ref 14 is that lentiviruses preferentially integrate into transcribed regions via their LEDGF/p75 PWWP domains (e.g. Gijsbers et al., JBC 2011). This might confound both CRE activity and barcode quantification.

There certainly could be biases associated with this, as with any other methods, which is in part why we conceived this study to cross-validate MPRA and transgenic assays. Independent methods pointing to the same result is one of the most powerful ways of finding true effects that we have in experimental sciences. Biased integration could contribute to poor reproducibility, but we speculate that variability in neuronal differentiation and cell culture probably prevailed in this case. The reason we think that, is other lentiMPRA carried out in established cell lines such as HepG2 provide Pearson replicate correlations way above 0.9 (see for example PMIDs 33046894 and 39814889). In PMID 33046894 we also compared the same library in HepG2 cells using 9 different MPRA techniques, including lentiMPRA, and found them to show good correlations between each other (see Figure 2b). In addition, lentiMPRA applied to differentiated ESCs to neurons at different time points has a Pearson R of 0.977 averaged for all 7 time points (PMID 31631012) and another lentiMPRA in primary neuronal cell types yielded Pearson correlation coefficients of 0.87-0.94 and in cerebral organoids - 0.89-0.90 (PMID:38781390). Another MPRA we conducted on the same WTC11 cell line (manuscript in preparation) also had better correlation coefficients (~0.83 vs this one's ~0.77), showing we can optimize the experimental setup to remove at least some of the undesired variation. In addition, it is worth noting that we use antirepressors on either side of the construct to protect from site of integration effects and that our median barcode count per sequence here is 103, thus averaging numerous integration sites and barcodes per element. Finally, we should point out PMID 30451991 from Barak Cohen's lab that tested the same MPRA library in different integration sites in the genome and found that while the genome location affected regulatory element activity, the relative strengths of the sequences were similar no matter the integration site. In our approach we average out the signal across numerous integration sites for each assayed sequence.

Reviewer #4

In this study Kosicki et al. conduct a lenti-viral-based massively parallel reporter assay in iPSC-derived neurons. Over 50,000 sequences were assayed, spanning several different sequence groupings, such as putative brain, heart, and kidney enhancers. In addition to wildtype sequences, synthetic and disease associated variants were also tested. Neuronal enhancers had elevated activity in the MPRA compared to other sequence groups, and within sequences derived from VISTA elements tested in mouse transgenic assays, the MPRA activity correlated with transgenic assay activity. In addition, the authors find that variants that alter MPRA activity also affected neuronal enhancer activity in mouse transgenic assays. Collectively, these results support the idea that for at least certain cell types, MPRA conducted in vitro can be used to functionally filter large numbers of sequences, to identify and analyze tissue specific enhancers. This work is significant as it addresses a currently understudied area - the correlation between MPRA data derived from short sequences tested in vitro and mouse transgenic assays, which remain the gold standard due to their ability to generate in vivo spatio-temporal data from full length enhancers. Overall, the study provides valuable context for the interpretation of MPRA, and supports their value as a high-throughput tool. There are, however, several areas that need clarification.

Thank you for the many thoughtful comments, we hope addressing them will clarify the message of this article for the readers.

Line 280 "We prioritized variants with links to important neuronal genes ..." This appears to have led to selection of elements for the transgenic assay that had low activity in the MPRA. For example, the activity of the reference element for hs268.1 was lower than the average negative control element, and the variant was expected to have even lower activity.

In the revised article, we improved our description of variant selection as follows: "We selected five of these variants for follow up in the transgenic assay, based on prior evidence of neuronal activity in transgenic assays and, to a lesser degree, links to important neuronal genes predicted using ABC model (e.g. QKI, PRKN, COA7 and MEF2C; Table 1, Figure 4a)"

The choice of reference elements with prior activity (e.g. hs268) was motivated by the desire to focus on variant effects - loss-of-function variants cannot be assessed in inactive elements. We were not trying to find novel active elements.

Indeed, of the seven variants tested, the three with a high baseline level of activity (>3) all had at least some loss of expression, while only one of the four with low baseline activity (<2) had a loss of expression.

Given that figure 3 showed that model validation rate increased with increasing MPRA activity, and a large part of the rationale for the study is to demonstrate the usefulness of using MPRA as a variant filter for transgenic mouse assays, the decision to select these low baseline activity elements for the transgenic assay is puzzling.

The model in Figure 3 does not predict the likelihood of *variant* impact, but of an element being positive for neural expression in the transgenic assay. In order to directly address the question of whether variants with significant impact on MPRA also affect the transgenic assay, we primarily selected our reference elements based on prior evidence of neural activity in the transgenic assay, not on whether these elements would have been predicted to be positive in the transgenic assay based on our MPRA model. We have added the following text to clarify this:

"We selected five of these variants for follow up in the transgenic assay, based on prior evidence of neuronal activity in transgenic assays and, to a lesser degree, links to important neuronal genes predicted using the ABC model (e.g. *QKI*, *PRKN*, *COA7* and *MEF2C*; Table 1, Figure 4a)⁵⁰."

Having said that, it is possible that variants with significant loss-of-function impact in tiles with high MPRA activity are more likely to affect activity in the transgenic assay than variants with significant loss-of-function impact in tiles with low MPRA activity, even controlling for their reference elements having similar baseline transgenic assay activity. Testing this idea would require many transgenic assays.

Additional elements with higher MPRA baseline activity should be tested.

We have now validated the predictive value of our general linear model (GLM) as follows: "To validate the model, we selected 6 elements with average predicted likelihood of *in vivo* neuronal activity of 84% and tested them using transgenic mouse assays. We found that 5 out of 6 such elements (86%) were active in neuronal tissues, showing that the model is well calibrated (Supplementary Figure 4).".

No follow up experiments were done to validate findings of the MPRA. For example, 13 motifs were observed to be enriched in high activity elements; a follow up MPRA experiment featuring ablation of these motifs, or synthetic elements with different combinations of the enriched motifs would reveal insights into neural transcriptional grammar.

This is a great suggestion but one that we feel is a bit outside the scope of this article. The goal of this study was not to learn about neuronal grammar, but to compare MPRA and transgenic assays. Therefore, Figure 2 and the associated text should be seen as QC and general description of the library. Most TFBS motifs found to be enriched in activators are already well known to be involved in neuronal activity (*LHX*, *DLX*, *CUX*, *RFX* families), which validates MPRA activity measures. We rephrased the revised article to stress QC is the main goal of this section:

"We then set out to analyze the transcription factor binding sites (TFBS) *to further validate our MPRA captures neuron-specific signals*."

We also note that we have a previous publication using MPRA for motif perturbation in neuronal stem cells (Kreimer 2022 PMID: 35315433)

Building synthetic elements is an exciting idea, one we previously tackled in other publications focusing on hepatocytes, due to their grammar being a bit more simplified with less 'words' i.e. TFs that regulate them (see PMIDs 23892608, 35573901, 37087538). We also note that sheer motif enrichment analysis is not enough to learn enhancer grammar, i.e. answer questions like how many motifs or in which combinations are sufficient to drive neuronal expression. Other groups have recently attempted to answer these questions using machine learning methods, which could potentially be more suited for this task (e.g. Taskiran 2024 PMID:38086419 or Gosai 2023 PMID:37609287). While our MPRA did capture neuron-specific signal, the amount of variability makes it unlikely we captured sufficient amounts to attempt building such a model.

Generation of new transgenics was limited to variants that caused loss of function in the MPRA. The impact of the study would be enriched by showcasing the ability of MPRA to find sequences that would be excluded from transgenic creation if solely non-functional filtering criteria were used (e.g. excluded if the region didn't have a classical enhancer chromatin signature). I would not require it, but making some transgenics from novel sequences that showed robust MPRA activity would significantly enhance the study.

We were not certain what the reviewer is suggesting here. Using transgenic assays to test reference elements with no active chromatin signatures, but high MPRA activity? Testing non-genomic elements (synthetic) with high MPRA activity? Or elements predicted by the MPRA to become active after variant introduction (gain-of-function mutations)? These are all great ideas and we have tested the last one. We identified an element with a relatively low reference MPRA activity (0.3) and a large gain-of-activity upon variant introduction (+8.3), and tested both alleles using the transgenic assay. While the reference element (hs3117) showed activity in the forebrain in the transgenic assay, the variant induced novel activity in the midbrain and the hindbrain. If we relied solely on reference activity measurements, we would likely not have tested this element using the transgenic assay.

Housekeeping promoters were highly active in the MPRA, and it's suggested that they can be used as universal positive controls in other MPRA. It's important to note that since they are cloned upstream of the minimal promoter they will function as normal promoter elements, directly driving expression of the reporter gene (this would not be the case in a different MPRA format such as STARR-seq). Thus, their high activity is probably not due to enhancer activity. This could be tested by looking at the orientation that promoters were inserted, and comparing the activity of promoters cloned in the forward orientation to promoters cloned in the reverse orientation, which presumably could not directly initiate transcription of the reporter gene.

We agree that it's likely. Two different MPRA articles from our lab in which tiles with the same coordinates were cloned in both orientations, revealed that promoter-derived tiles are statistically more likely to have asymmetrical activity (PMID:33046894 and 39814889). For the latter (PMID 39814889) we tested over 19,000 promoters in either orientation finding a stronger orientation effect for promoters versus enhancers. Our results in that study were comparable to previous results not using a 32bp minimal promoter that was used both there and here and the

motif enrichments for promoters were also comparable. We added "We note that they [housekeeping promoter tiles] may function as autonomous promoters and not as enhancers." We also agree that these could be great 'universal' controls for MPRA and as part of the IGVF consortium are working to use them for that.

Supplementary tables 2-4 were not included in the reviewer file.

We apologize for this. It could be the fault of the outdated manuscript submission system. The original XLSX file (which should be available to the reviewer) contains all tables as tabs. The manuscript submission system converts everything to a PDF file that includes only the first tab of the XLSX file, and breaks any table wider than a page into page-sized pieces. We included a note to editor to point the reviewers to the original Supplementary Tables file directly to avoid this in the future as even splitting the XLSX file into many would not fix the width issue.

Line 190 – It would be nice to include images of the motifs for the 13 motifs found to be enriched, as done in Fig4c,d. Adding de novo motif discovery to the analysis would also be interesting.

We have now added de novo motif analysis and following text:

"Repressor tiles were enriched for Nkx6.1 and four motifs from the SOX family^{46,47}, but only in analysis using genomic background. This may imply lack of specific repressive signal in our library, limited power to detect such signal or relative dearth of known repressive motifs in the the HOMER dataset. The latter might be consistent with de novo motif analysis conducted against tiles with background activity, which revealed similar motifs for activator tiles (RFX7/Rfxdc2, X-box, SP5, match scores 0.82-0.9), but novel repressor motifs with tentative matches to ZIC2, KLF15, and SP1 among others (match scores of 0.7-0.75; Supplementary Table 3)."

We also added images of RFX, SP5, SOX6 and ATF7 to Supplementary Figure 2 and added raw results of HOMER runs, including motif images, as Supplementary Data 1.

In Figure 4a the variant tested by transgenic assay is colored in blue, which is difficult to see. Consider a more distinct color.

Changed to orange.

The epigenetic signal is described as coming from "WTC11 datasets" on line 211, but then described as "differentiated WTC11 cells" on line 221. I guess they are both referring to differentiated WTC11 cells, but labelling this group as only WTC11 on the graph in Fig. 2d is confusing. I would suggest changing it to "WTC11 Neurons" or something similar.

That's a good suggestion, now updated.

The panel references in the text for Figure 2 appear to be incorrect. For example, it looks like line 216 Fig 2b should actually be referencing Fig. 2d. “Fig. 2d left” on line 221 should be “Fig. 2d right”.

Thanks for catching this! It has been corrected.

Reviewer #5

Thanks for helping review this and the helpful comments!

We would like to thank all the reviewers for their helpful comments!

Reviewer #1 (Remarks to the Author):

The manuscript has been revised to address all the concerns I raised in the initial review. Furthermore, improved statistical analyses were employed that handled data variability more thoroughly and implemented a multiple-testing correction, which reduced the number of tiles designated as activators and repressors. This improved analysis, along with several additional experiments and analyses, strengthened this manuscript.

Signing this review,
Ross Hardison

Reviewer #2 (Remarks to the Author):

Major and minor comments provided to authors have been addressed or, at minimum, clarified within the manuscript. This has enhanced readability. Figures added to supplement has also been included, adding further understanding of the underlying methodologies utilized within this project.

Reviewer #3 (Remarks to the Author):

The authors have provided answers to our questions and concerns. We have no further comments.

Reviewer #4 (Remarks to the Author):

This revision features improvements in several aspects of the analysis, including improved barcode handling, resulting in less variability between MPRA replicates, a more robust motif analysis, and additional comparisons with pre-existing MPRA datasets. Additional transgenic data was also added, and several areas in the original manuscript that were confusing have received clarifying text and are now significantly improved. I still think that basing candidate selection for the variant transgenic assay on evidence of neuronal activity in transgenic assays and links to neuronal genes, rather than focusing on the MPRA activity, is a missed opportunity. However, I am sympathetic to the explanation given and find that the revisions to this section of the manuscript are a substantial improvement. I believe my critiques have been sufficiently addressed, and have only one minor comment regarding line 288: “we selected 6 elements with an average predicted likelihood of *in vivo* neuronal activity of 86% and tested them using transgenic mouse assays.” It would be helpful to add the MPRA activity of each tested element to Supplemental Figure 4.

We have added MPRA activity (and predicted likelihood of *in vivo* neuronal activity) to the figure.

Reviewer #5 (Remarks to the Author):

I co-reviewed this manuscript with one of the reviewers who provided the listed reports.

This is part of the Nature Communications initiative to facilitate training in peer review and to provide appropriate recognition for Early Career Researchers who co-review manuscripts.